# Mosaic cis-regulatory evolution drives transcriptional partitioning of HERVH endogenous retrovirus in the human embryo

Thomas A Carter[1], Manvendra Singh[1], Gabrijela Dumbović[2,3], Jason D Chobirko[1], John L Rinn[2,4], Cédric Feschotte[1]*

[1]Department of Molecular Biology and Genetics, Cornell University, Ithaca, United States; [2]BioFrontiers Institute, University of Colorado Boulder, Boulder, United States; [3]Institute for Cardiovascular Regeneration, Goethe University Frankfurt, Frankfurt am Main, Germany; [4]Department of Biochemistry, University of Colorado Boulder, Boulder, United States

**Abstract** The human endogenous retrovirus type-H (HERVH) family is expressed in the preimplantation embryo. A subset of these elements are specifically transcribed in pluripotent stem cells where they appear to exert regulatory activities promoting self-renewal and pluripotency. How HERVH elements achieve such transcriptional specificity remains poorly understood. To uncover the sequence features underlying HERVH transcriptional activity, we performed a phyloregulatory analysis of the long terminal repeats (LTR7) of the HERVH family, which harbor its promoter, using a wealth of regulatory genomics data. We found that the family includes at least eight previously unrecognized subfamilies that have been active at different timepoints in primate evolution and display distinct expression patterns during human embryonic development. Notably, nearly all HERVH elements transcribed in ESCs belong to one of the youngest subfamilies we dubbed LTR7up. LTR7 sequence evolution was driven by a mixture of mutational processes, including point mutations, duplications, and multiple recombination events between subfamilies, that led to transcription factor binding motif modules characteristic of each subfamily. Using a reporter assay, we show that one such motif, a predicted SOX2/3 binding site unique to LTR7up, is essential for robust promoter activity in induced pluripotent stem cells. Together these findings illuminate the mechanisms by which HERVH diversified its expression pattern during evolution to colonize distinct cellular niches within the human embryo.

*For correspondence:
cf458@cornell.edu

**Competing interest:** The authors declare that no competing interests exist.

## Editor's evaluation

Transposons achieve evolutionary success only upon self-replication in the germline, where novel genomic insertions are passed to the next generation. To define the genetic determinants of transposon specialization to the germline and specifically, specialization to pluripotent embryonic stem cells, this article applies evolutionary analysis, cell type-specific transcriptomics, and expression reporter assays to the human endogenous retrovirus type-H, HERV. Previous links between HERV and the maintenance of pluripotency make the discoveries consequential for the mobile element and genome evolution communities along with those engaged in stem cell biology and regenerative medicine.

## Introduction

Transposable elements (TEs) are genomic parasites that use the host cell machinery for their own propagation. To propagate in the host genome, they must generate new insertions in germ cells or their embryonic precursors, as to be passed on to the next generation (*Charlesworth and Langley, 1986*; *Cosby et al., 2019*; *Haig, 2016*). To this end, many TEs have evolved stage-specific expression in germ cells or early embryonic development (*Faulkner et al., 2009*; *Fort et al., 2014*; *Göke et al., 2015*; *Miao et al., 2020*; *Urusov et al., 2011*). But how does this precise control of TE expression evolve?

Many endogenous retroviruses (ERVs) are known to exhibit highly stage-specific expression during early embryonic development (*Chang et al., 2021*; *Göke et al., 2015*; *Hermant and Torres-Padilla, 2021*; *Peaston et al., 2004*; *Svoboda et al., 2004*). ERVs are derived from exogenous retroviruses with which they share the same prototypical structure with two long terminal repeats (LTRs) flanking an internal region encoding products promoting their replication (*Eickbush and Malik, 2002*). There are hundreds of ERV families and subfamilies in the human genome, each associated to unique LTR sequences (*Kojima, 2018*; *Vargiu et al., 2016*). Each family has infiltrated the germline at different evolutionary timepoints and have achieved various levels of genomic amplification (*Bannert and Kurth, 2004*; *Vargiu et al., 2016*). One of the most abundant families is HERVH, a family derived from a gamma retrovirus that first entered the genome of the common ancestor of apes, Old World monkeys (OWMs), and New World monkeys more than 40 million years ago (mya) (*Goodchild et al., 1993*; *Izsvák et al., 2016*; *Mager and Freeman, 1995*). To date, no polymorphic HERVH insertions have been found in the human genome (*Thomas et al., 2018*).

There are four subfamilies of HERVH elements currently recognized in the Dfam (*Storer et al., 2021*) and Repbase (*Bao et al., 2015*; *Kojima, 2018*) databases and annotated in the reference human genome based on distinct LTR consensus sequences: LTR7 (formerly known as Type I), 7b (Type II), 7c, and 7y (Type Ia) (*Bao et al., 2015*; *Goodchild et al., 1993*; *Jern et al., 2005*; *Jern et al., 2004*). Additional subdivisions of HERVH elements were also proposed based on phylogenetic analysis and structural variation of their internal gene sequences (*Gemmell et al., 2019*; *Jern et al., 2005*; *Jern et al., 2004*). However, all HERVH elements are currently annotated in the human genome using a single consensus sequence for the internal region (HERVH_int) and the aforementioned four LTR subfamilies.

HERVH has been the focus of extensive genomic investigation for its high level of RNA expression in human embryonic stem cells (ESCs) and induced pluripotent stem cells (iPSCs) (*Fort et al., 2014*; *Gemmell et al., 2015*; *Izsvák et al., 2016*; *Kelley and Rinn, 2012*; *Loewer et al., 2010*; *Römer et al., 2017*; *Santoni et al., 2012*; *Zhang et al., 2019*). Several studies showed that family-wide HERVH knockdown results in the loss of pluripotency of human ESC and reduced reprogramming efficiency of somatic cells to iPSC (*Lu et al., 2014*; *Ohnuki et al., 2014*; *Wang et al., 2014*). Others reported similar phenotypes with the knockdown of individual HERVH-derived RNAs such as those produced from the *lincRNA-RoR* and *ESRG* loci (*Loewer et al., 2010*; *Wang et al., 2014*) or the deletion of individual HERVH loci acting as boundaries for topological associated domains (*Zhang et al., 2019*). These results converge on the notion that HERVH products (RNA or proteins) exert some modulatory effect on the cellular homeostasis of pluripotent stem cells. However, it is important to emphasize that different HERVH knockdown constructs produced variable results and inconsistent phenotypes (*Lu et al., 2014*; *Wang et al., 2014*; *Zhang et al., 2019*), and a recent knockout experiment of the most highly transcribed locus (*ESRG*) failed to recapitulate its previous knockdown phenotype (*Takahashi et al., 2021*). These inconsistent results have failed to clarify which expressed HERVH loci, if any, are necessary for the maintenance of pluripotency.

The mechanisms regulating the transcription of HERVH also remain poorly understood. RNA-seq analyses have established that HERVH expression in human ESCs, iPSCs, and the pluripotent epiblast can be attributed to a relatively small subset of loci (estimated between 83 and 209) driven by LTR7 (sensu stricto) sequences (*Göke et al., 2015*; *Wang et al., 2014*; *Zhang et al., 2019*). The related 7y sequences are known to be expressed in the pluripotent epiblast of human embryos (*Göke et al., 2015*) and a distinct subset of elements associated with 7b and 7y sequences are expressed even earlier in development at the onset of embryonic genome activation (*Göke et al., 2015*). These observations suggest that the HERVH family is composed of subsets of elements expressed at different timepoints during embryonic development and that these expression patterns reflect, at least in part,

the unique cis-regulatory activities of their LTRs. While it has been reported that several transcription factors (TFs) bind and activate HERVH LTRs, including the pluripotency factors OCT4, NANOG, SP1, and SOX2 (*Göke et al., 2015*; *Ito et al., 2017*; *Kelley and Rinn, 2012*; *Kunarso et al., 2010*; *Ohnuki et al., 2014*; *Pontis et al., 2019*; *Santoni et al., 2012*), it remains unclear how TF binding contributes to the differential expression of HERVH subfamilies and why only a minority of HERVH are robustly transcribed in pluripotent stem cells and embryonic development.

To shed light on these questions, we focused this study on the cis-regulatory evolution of LTR7 elements. We use a 'phyloregulatory' approach combining phylogenetic analyses and regulatory genomics to investigate the sequence determinants underlying the partitioning of expression of HERVH/LTR7 subfamilies during early embryonic development.

## Results

### LTR7 consists of eight previously undefined subfamilies

We began our investigation by examining the sequence relationships of the four LTR7 subfamilies currently recognized in the human genome: LTR7 sensu stricto (748 proviral copies; 711 solo LTRs), 7b (113; 524), 7c (24; 223), and 7y (77; 77). We built a maximum likelihood phylogenetic tree from a multiple sequence alignment of a total of 781 5′ LTR and 1073 solo LTR sequences of near-complete length (>350 bp) representing all intact LTR subfamilies extracted from the RepeatMasker output of the hg38 human reference assembly. While 7b and 7y sequences cluster, as expected, into clear monophyletic clades with relatively short internode distances and little subclade structure, sequences from the 7c and LTR7 subfamilies were much more heterogeneous and formed many subclades (*Figure 1A*). Notably, sequences annotated as LTR7 were split into distinct monophyletic clades indicative of previously unrecognized subfamilies within that group. The branch length separating some of these LTR7 subclades were longer from one another than they were from those falling within the 7b, 7c, and 7y clades, indicating that they represent subfamilies as different from each other as those previously recognized (*Figure 1A*).

We next sought to classify LTR7 elements more finely by performing a phylogenetic analysis using a multiple sequence alignment of all intact LTR7 sequences (>350 bp) along with the consensus sequences for the other LTR7 subfamilies for reference. We defined high-confidence subfamilies as those forming a clade supported by >95% ultrafast bootstrap (UFbootstrap) and internal branches > 0.015 (1.5 nucleotide substitutions per 100 bp) separating subgroup nodes. Based on these criteria, LTR7 elements could be divided into eight subfamilies (*Figure 1B*).

While long internal branches with high UFbootstrap support separate LTR7 subfamilies, intra-subfamily internal branches with >95% UFbootstrap support were shorter (<0.015), suggesting that each subfamily was the product of a rapid burst of amplification of a common ancestor. To approximate the sequence of these ancestral elements, we generated majority rule consensus sequence for each of the eight newly defined LTR7 subfamilies (7o, 7bc, 7up, etc.). The consensus sequences were deposited at https://www.dfam.org.

To investigate the evolutionary relationships among the newly defined and previously known LTR7 subfamilies, we conducted a median-joining network analysis (*Leigh and Bryant, 2015*) of their consensus sequences (*Figure 1C*). The network analysis provides additional information on the relationships between subfamilies and approximates the shortest and most parsimonious paths between them (*Bandelt et al., 1999*; *Cordaux et al., 2004*; *Posada and Crandall, 2001*). The results place 7o in a central position from which two major lineages are derived. One lineage led to two sub-lineages, formed by 7up1, 7up2, and 7u1 (with 7up1 and 7up2 being most closely related) and by 7d1 and 7d2. The other lineage emanating from 7o rapidly split into two sub-lineages; one gave rise to 7u2 and then to 7y and the other gave rise to 7bc which is connected to the two more diverged subfamilies 7b and 7c (*Figure 1C*). Together these results indicate that the LTRs of HERVH elements can be divided into additional subfamilies than those previously recognized.

### The age of LTR7 subfamilies suggests three major waves of HERVH propagation

The genetic differences between LTR7 subfamilies suggest that they may have been active at different evolutionary timepoints. To examine this, we used reciprocal *liftover* analysis to infer the presence/

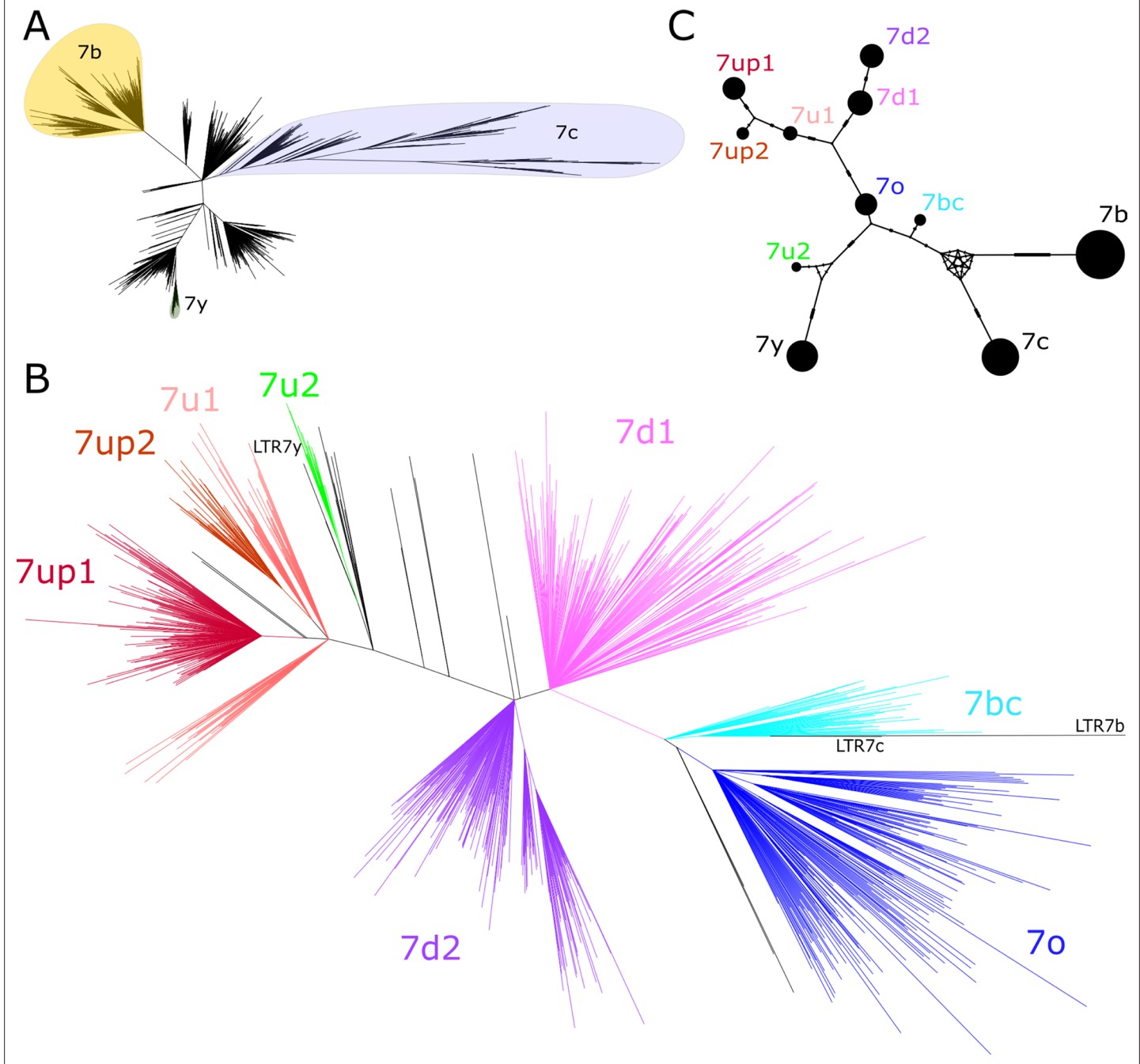

**Figure 1.** Phylogenetic analysis of LTR7 sequences. (**A**) Unrooted phylogeny of all solo and 5′ LTR7 subfamilies from LTR7, 7b, 7c, and 7y. Colors denote clades consisting of previously annotated 7b, 7c, and 7y with >95% concordance. (**B**) Unrooted phylogeny of all solo and 5′ LTR7 sequences. All nodes with ultrafast bootstraps (UFbootstraps) > 0.95, >10 member insertions, and >1.5 substitutions/100 bp (~6 base pairs) are grouped and colored (see Materials and methods). Previously listed consensus sequences from 7b/c/y were included in the alignment and are shown in black. (**C**) Median-joining network analysis of all LTR7 and related majority rule consensus sequences. Ticks indicate the number of SNPs at non-gaps between consensus sequences. The size of circles is proportional to the number of members in each subfamily. Only LTR7 insertions that met filtering requirements (see Materials and methods) are included while 7b/c/y counts are from dfam.

The online version of this article includes the following figure supplement(s) for figure 1:

**Figure supplement 1.** Phylogenetic tree from LTR7 reverse transcriptase (RVT) domains.

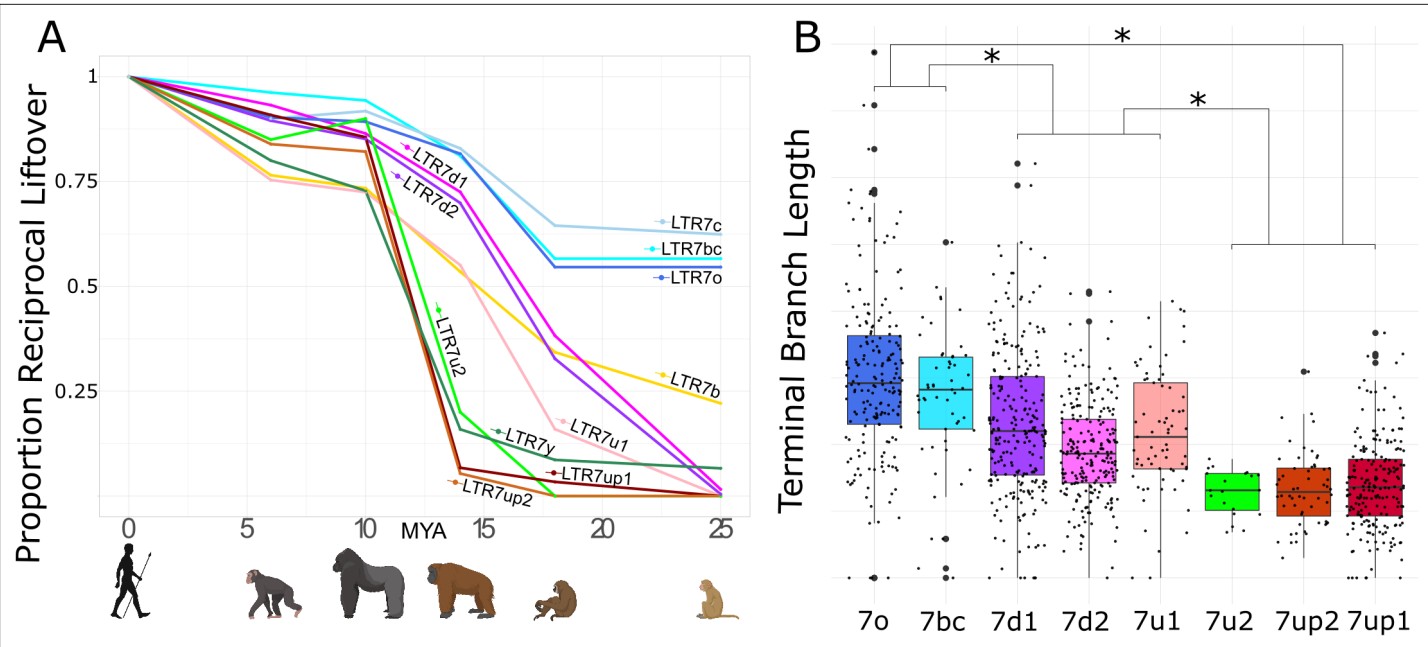

**Figure 2.** Age analysis of LTR7 subfamilies. (**A**) Proportion of a given subfamily that have 1:1 orthologous insertions between human and other primate species. LTR7 subfamilies are from the tree in **Figure 1B**; 7b/c/y subfamilies are from RepeatMasker annotations. Non-human primates are spaced out on the X axis in accordance with their approximate divergence times to the human lineage. (**B**) Terminal branch lengths of all LTR7 insertions from **Figure 1B**. Groups with similar liftover profiles were merged for statistical testing (see Materials and methods). Differences with padj <1e-15 are denoted with * (Wilcoxon rank-sum test with Bonferroni correction).

The online version of this article includes the following figure supplement(s) for figure 2:

**Figure supplement 1.** Gross reciprocal liftover from human to non-human primate (NHP).

absence of each human LTR7 locus across five other primate genomes. Insertions shared at orthologous genomic position across a set of species are deemed to be ancestral to these species and thus can be inferred to be at least as old as the divergence time of these species and likely fixed within each species (**Johnson, 2019**).

The results of this cross-species analysis indicate that LTR7 subfamilies have been transpositionally active at different timepoints in the primate lineage (**Figure 2A**). The subfamilies 7o, 7bc, and 7c are the oldest since the majority of their insertions are found at orthologous position in rhesus macaque, an OWM. These three subfamilies share similar evolutionary trajectories, with most of their proliferation occurring prior to the split of OWM and hominoids, ~25 mya (**Figure 2A**). Members of the 7b subfamily (the most numerous, 637 solo and full-length insertions) appear to be overall younger, since only 22% of the human 7b elements could be lifted over to rhesus macaque and the vast majority appeared to have inserted between 10 and 20 mya (**Figure 2A**, **Figure 2—figure supplement 1**). Only 5 of the 550 elements in the 7d1 and 7d2 subfamilies could be retrieved in rhesus macaque, but ~30% were shared with gibbon and ~75% were shared with orangutan. Thus, these two subfamilies are largely hominoid-specific and achieved most of their proliferation prior to the split of African and Asian great apes ~14 mya (**Figure 2A**). Members of the 7u1 subfamily also emerged in the hominoid ancestor, but the majority (55%) of 7u1 elements present in the human genome inserted after the split of gibbons in the great ape ancestor, between 14 and 20 mya. Thus, the 7b, 7d1/2, and 7u1 subfamilies primarily amplified during the same evolutionary window, 14–20 mya.

The 7up1/2, 7y, and 7u2 subfamilies represent the youngest in the human genome, with most of their proliferation occurring between ~10 and ~ 14 mya, in the ancestor of African great apes (**Figure 2A**). Based on these results, these subfamilies seem to have experienced a burst of transposition after the divergence of African and Asian great apes but before the split of the pan/homo and gorilla lineages. For example, only 14 of the 208 (6.7%) human 7up1 elements can be retrieved in orangutan, but 178 (85.6%) can be found in gorilla. These data indicate that the three youngest LTR7 subfamilies mostly expanded in the ancestor of African great apes (**Figure 2A**).

As an independent dating method, we used the terminal branch length separating each insertion from its nearest node in *Figure 1B* (*Figure 2*). Here, the terminal branch lengths are proportional to nucleotide divergence accumulated after insertion and can thus approximate each insertion's relative age. This method largely corroborated the results of the *liftover* analysis and revealed three age groups among LTR7 subfamilies characterized by statistically different mean branch lengths (p(adj) < 1e-15; Wilcoxon rank-sum test). By contrast, we found no statistical difference between the mean branch length of the subfamilies within these three age groups, suggesting that they were concomitantly active. Taken together, our dating analyses distinguish three major waves of HERV propagation: an older wave 25–40 mya involving 7c, 7o, and 7bc elements, an intermediate wave 9–20 mya involving 7b, 7d1/2, and 7u1, and a most recent wave 4–10 mya implicating primarily 7up1/2, 7u2, and 7y elements.

## Only LTR7up shows robust transcription in human ESC and iPSC

Our data thus far indicate that LTR7 is composed of genetically and evolutionarily distinct subfamilies. Because a subset of HERVH elements linked to LTR7 were previously reported to be transcribed in pluripotent stem cells (human ESCs and iPSCs), we wondered whether this activity was restricted to one or several of the LTR7 subfamilies newly defined herein. To investigate this, we performed a 'phyloregulatory' analysis, where we layered locus-specific regulatory data obtained from publicly available genome-wide assays in ESCs (mostly from the H1 cell line, see Materials and methods) for each LTR insertion on top of a phylogenetic tree depicting their evolutionary relationship. We called an individual LTR7 insertion as positive for a given feature if there is overlap between the coordinates of the LTR and that of a peak called for this mark (see Materials and methods). We predicted that if transcriptional activity was an ancestral property of a given subfamily, evidence of transcription and 'activation' marks should be clustered within the cognate clade. Alternatively, if transcription and activation marks were to be distributed throughout the tree, it would indicate that LTR7 transcriptional activity in pluripotent cells was primarily driven by post-insertional sequence divergence or context-specific effects such as local chromatin or cis-regulatory environments. Differences in the proportion of positive insertions for a given mark between LTR7 subfamilies were tested using a chi-square test with Bonferroni correction. Unless otherwise noted, all proportions compared thereafter were significantly different (padj < 0.05).

The results (*Figure 3A*) show that HERVH elements inferred to be 'highly expressed' (fpkm > 2) based on RNA-seq analysis (*Wang et al., 2014*) were largely confined to two closely related subfamilies, 7up1 and 7up2, together referred to as 7up hereafter. Indeed, we estimated that 33% of 7up elements (88 loci) are highly expressed according to RNA-seq compared with only 2% of highly expressed elements from all other subfamilies combined (17 loci). Nascent RNA mapping using GRO-seq data (*Estarás et al., 2015*) recapitulated this trend with 22% of 7up loci with visible signal (*Figure 3—figure supplement 1*), compared with only 4% of other LTR7 loci (*Figure 3D*, *Figure 3—figure supplement 1*). Half of the loci displaying GRO-seq signal (53/96) also showed evidence of mature RNA product (*Supplementary file 1*). Thus, HERVH transcriptional activity in H1 ESCs is largely limited to loci driven by 7up sequences.

As previously noted from ChIP-seq data (*Ohnuki et al., 2014*), we found that KLF4 binding is a strong predictor of transcriptional activity: KLF4 ChIP-seq peaks overlap 91% of 7up loci and KLF4 binding is strongly enriched for the 7up subfamilies relative to other subfamilies (*Figure 3A, B, and D*). NANOG binding is also enriched for 7up (97.7% of loci overlap ChIP-seq peaks) but is observed to varying degrees at other LTR7 loci that do not show evidence of active transcription based on GRO-seq and/or RNA-seq (85% of 7u1 loci, 32% 7d1, 45% 7d2, 13% 7o, 8.7% 7bc, and 0% of 7u2). While KLF4 and NANOG binding is pervasive across the 7up subfamily, OCT4 binds merely 12% of 7up loci. Other TFs with known roles in pluripotency are also enriched at 7up loci, such as SOX2 (32% LTR7up, 1–3% all other LTR7), FOXP1(49%, 0–4.3%), and FOXA1(28%, 0–1.4%). In fact, FOXA1 binds only a single non-7up insertion in our dataset, making it the most exclusive feature of 7up loci among the TFs examined in this analysis (see *Supplementary file 8* for full statistical analysis of all marks).

Congruent with having generally more TF binding and transcriptional activity, 7up loci also have a propensity to be decorated by H3K4me3, a mark of active promoters (76% LTR7up vs. 19% all others) and the broader activity mark H3K27ac (89% vs. 48%) (*Figure 3A, B*). By contrast, H3K4me1, a mark typically associated with low POLII loading as seen at enhancers as opposed to promoters, is spread rather evenly

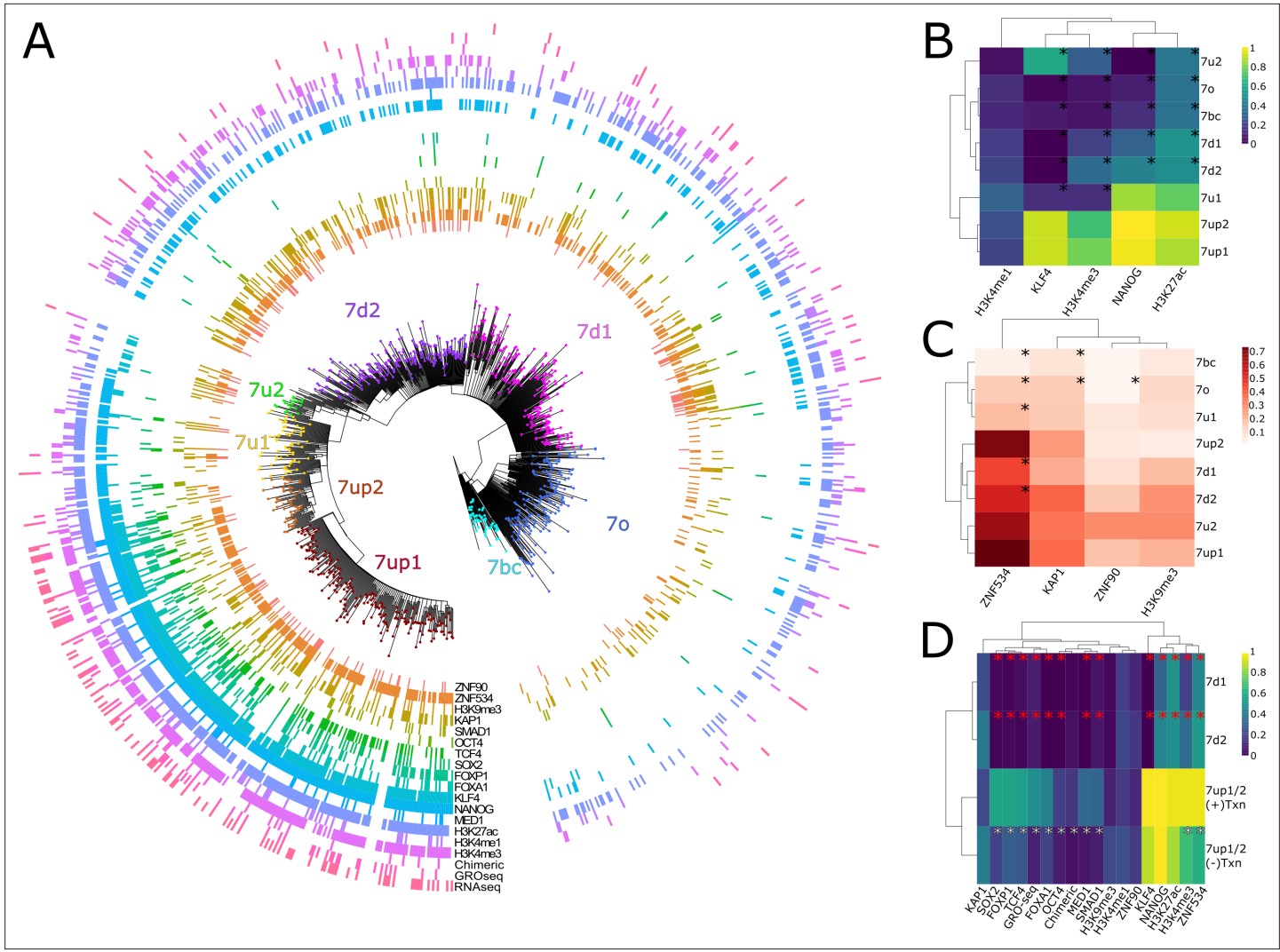

**Figure 3.** Phyloregulatory analysis of LTR7. (**A**) '"Phyloregulatory' map of LTR7. The phylogenetic analysis to derive the circular tree is the same as for the tree in **Figure 1B** but rooted on the 7b consensus. Subfamilies defined in **Figure 1** are denoted with dotted colored tips. Positive regulatory calls for each insertion are shown as tick marks of different colors and no tick mark indicates a negative call. All marks are derived from embryonic stem cell (ESC) except for ZNF90 and ZNF534, which are derived from ChIP-exo data after overexpression of these factors in HEK293 cells (see Materials and methods). (**B**) Heatmap of major activation and repression profiles. Proportions indicate the proportion of each group positive for a given characteristic. Trees group LTR7 subfamilies on regulatory signature, not sequence similarity. Asterisks denote statistical differences between given group and 7up1 (padj < 0.05 Wilcoxon rank-sum with Bonferroni correction). (**C**) Heatmap done in similar fashion to **B** but for repression marks. (**D**) Heatmap of transcribed (>2 fpkm) and untranscribed 7up1/2 (<2 fpkm) and all 7d1/2. Red asterisks denote statistical differences between 7d1/2 and 7up1 (padj < 0.05 chi-square Bonferroni correction). White asterisks denote differences between transcribed and untranscribed LTR7up.

The online version of this article includes the following figure supplement(s) for figure 3:

**Figure supplement 1.** Heatmap and aggregate signal of two replicates of whole-genome bisulfite sequencing (WGBS), GRO-seq (plus strand), H3K9me3, and SOX2 in H1 cells.

**Figure supplement 2.** Violin plots visualize the density and distribution of ZNF534 in early embryogenesis and ESCs at passages 0 and 10.

throughout the tree of LTR7 sequences (26% vs. 18%) (**Figure 3A, B**). Thus, promoter marks are primarily restricted to 7up loci, but a broader range of LTR7 loci display putative enhancer marks.

Taken together, our phyloregulatory analysis suggests that strong promoter activity in ESCs is restricted to 7up elements.

## Differential activation, rather than repression, explain the differential transcriptional activity of LTR7 subfamilies in ESCs

The pattern described above could be explained by two non-mutually exclusive hypotheses: (i) 7up elements (most likely their progenitor) have acquired unique sequences (TF binding sites, TFBS) that promote Pol II recruitment and active transcription, and/or (ii) they somehow escape repressive mechanisms that actively target the other subfamilies, preventing their transcription. For instance, 7up elements may lack sequences targeted by transcriptional repressors such as KRAB-zinc finger proteins (KZFPs) that silence the other subfamilies in ESCs. KZFPs are well known for binding TEs in a subfamily-specific manner where they nucleate inheritable epigenetic silencing (*Ecco et al., 2017*; *Jacobs et al., 2014*; *Wolf et al., 2020*; *Yang et al., 2017*) and several KZFPs are known to be capable of binding LTR7 loci (*Imbeault et al., 2017*). To examine whether KZFPs may differentially bind to LTR7 subfamilies, we analyzed the loading of the corepressor KAP1 and the repressive histone mark H3K9me3 typically deposited through the KZFP/KAP1 complex, across the LTR7 phylogeny using ChIP-seq data previously generated for ESCs (*Imbeault et al., 2017*; *Theunissen et al., 2016*). We found that KAP1 and H3K9me3 loading were neither enriched nor depleted for 7up elements relative to other subfamilies (*Figure 3A, C*). Overall, there were no significant differences in the level of H3K9me3 marking across subfamilies and the only difference in KAP1 binding was a slight but significant depletion for 7bc and 7o compared to all other subfamilies including 7up (14% vs. 35% – padj < 0.05 chi-square Bonferroni correction). Furthermore, KAP1 and H3K9me3 loading were found in similar proportions in expressed and unexpressed 7up elements (padj > 0.05) (*Figure 3C*). This was also the case for CpG methylation, whose presence was not differential between subfamilies (padj > 0.05 Wilcoxon rank-sum with Bonferroni correction) (*Figure 3—figure supplement 1*). Thus, KAP1 binding and repressive marks at LTR7 in ESCs poorly correlate with their transcriptional activity and differential repression is unlikely to explain the differential promoter activity of LTR7 subfamilies in ESCs.

We also examined the binding profile of ZNF534 and ZNF90, two KZFPs previously reported to be enriched for binding LTR7 elements using ChIP-exo data in human embryonic kidney 293 cells (*Imbeault et al., 2017*), in order to examine whether they bind a particular subset of elements in our LTR7 phylogeny. We found that while ZNF90 bound all LTR7 subfamilies to a similar extent, ZNF534 preferentially bound members of the 7up subfamily (72% of LTR7up vs. 34–53% of non-LTR7up). However, ZNF534 binding in 293 cells did not correlate with transcriptional activity of 7up elements in ESCs nor with KAP1 binding or H3K9me3 deposition in these cells (*Figure 3A, D*). In other words, there was no significant enrichment for ZNF534 binding within untranscribed 7up elements nor depletion within the 7up elements we inferred to be highly transcribed in ESCs. These observations could simply reflect the fact that ZNF534 itself is not highly expressed in ESCs (*Figure 3—figure supplement 2*) and do not preclude that ZNF534 represses 7up in other cellular contexts or cell types. Collectively, these data suggest that differential LTR binding of KZFP/KAP1 across subfamilies cannot readily explain their differential regulatory activities in ESCs. Thus, differential activation is the most likely driver for the promoter activity of 7up elements in ESCs.

To determine which factors are associated and potentially determinant for 7up promoter activity, we compared the set of 'highly expressed' 7up loci to 7up loci which are apparently poorly expressed, using 7d1/d2 as outgroups (*Figure 3D*). While known regulators of LTR7 transcription, KLF4 and NANOG, are enriched for binding to 7up elements, their occupancy alone cannot distinguish transcribed from untranscribed 7up loci (*Figure 3D*). Thus, other factors must contribute to the transcriptional activation of 7up elements. Our analysis of pluripotent transcriptional activators SOX2, FOXA1, FOXP1, OCT4, TCF4, and SMAD1 (*Boyer et al., 2005*; *Chambers and Smith, 2004*; *Niwa, 2007*) binding profiles show that all of these TFs are enriched in robustly transcribed 7up loci compared to non-transcribed loci (*Figure 3D*). Intriguingly, when overexpressed in HEK293 cells, the potential KZFP repressor ZNF534 preferably binds ESC-transcribed 7up over untranscribed 7up, suggesting that ZNF534 may suppress transcription-competent 7up in cellular contexts where this factor is expressed.

Together these data suggest that differential repression cannot explain the differential promoter activity of LTR7 subfamilies in ESCs but rather that highly expressed LTR7up loci are preferentially bound by a cocktail of transcriptional activators that are less prevalent on poorly expressed loci.

## Inter-element recombination and intra-element duplication drove LTR7 sequence evolution

The data presented above suggest that the transcriptional activity of 7up in ESCs emerged from the gain of a unique combination of TFBS. To identify sequences unique to 7up relative to its closely related subfamilies, we aligned the consensus sequences of the newly defined LTR7 subfamilies and those of 7b/c/y consensus sequences. This multiple sequence alignment revealed blocks of sequences that tend to be highly conserved across subfamilies, only diverging by a few SNPs, while other regions showed insertion/deletion (indel) segments specific to one or a few subfamilies (*Figure 4A*).

Upon closer scrutiny, we noticed that the indels characterizing some of the subfamilies were at odds with the evolutionary relationship of the subfamilies defined by overall phylogenetic and network analyses. This was particularly obvious in segments we termed block 2b (where 7y and 7u2 share a large insertion with 7b and 7c) and block 3 (where 7y and 7b share a large insertion). This led us to carefully examine the multiple sequence alignment of the LTR7 consensus sequences to identify indels with different patterns of inter-subfamily relationships. Based on this analysis, we defined seven sequence blocks shared by a different subset of subfamilies, pointing at relationships that were at odds with the overall phylogeny of the LTR7 subfamilies (*Figure 4A–B*). These observations suggested that some of the blocks have been exchanged between LTR7 subfamilies through recombination events.

To systematically test if recombination events between elements drove the evolution of LTR7 subfamilies, we generated parsimony trees for each block of consensus sequences and looked for incongruences with the overall consensus phylogeny. We found a minimum of six instances of clades supported in the block parsimony trees that were incongruent with those supported by the overall phylogeny (*Figure 4B and D*).

We also found some blocks evolved via tandem duplication. Notably, block 2b was absent from 7d1/2 and 7bc/o but present in all other subfamilies. However, block 2b from 7b, 7c, 7u2, and 7y aligned poorly with block 2b from 7up and 7u1. Instead, block 2b from 7up/u1 2b was closely related (~81% nucleotide similarity) to block 2a from the same subfamilies (*Figure 4C*, *Figure 4—figure supplement 1*), suggestive that it arose via tandem duplication in the common ancestor of these subfamilies.

The results above suggest that the evolution of HERVH was characterized by extensive diversification of LTR sequences through a mixture of point mutations, indels, and recombination events.

## HERVH subfamilies show distinct expression profiles in the preimplantation embryo

We hypothesized that the mosaic pattern of LTR sequence evolution, which represents a mixture of point mutations, duplications, and inter-element recombination, gave rise to TFBS combinations unique to each family that drove shifts in HERVH expression during early embryogenesis. To test this, we aimed to reanalyze the expression profiles of newly defined LTR7 subfamilies during early human embryogenesis and correlate these patterns with the acquisition of embryonic TF binding motifs within each of the subfamilies.

To perform this analysis, we first reannotated the hg38 reference genome assembly using Repeat-Masker with a custom library consisting of the consensus sequences for the eight newly defined LTR7 subfamilies plus newly generated consensus sequences for 7b, 7c, and 7y subfamilies redefined from the phylogenetic analysis presented in *Figure 1B* (*Figure 5—figure supplement 1*) (see Materials and methods). Our newly generated RepeatMasker annotations (*Supplementary file 2*) did not drastically differ from previous annotations of LTR7 and 7c, where 90% and 86% of insertions, respectively, were concordant with the old RepeatMasker annotations (though LTR7 insertions were now assigned to one of the eight newly defined subfamilies). 7y and 7b annotations, however, shifted significantly. Only 33% of previously annotated 7y reannotated concordantly with 53% now being annotated as 7u2 and only 52% of 7b reannotated concordantly, with 22% now annotated as 7y. These shifts can be largely explained by the fact that 7u2 and 7y are closely related (*Figure 1A–C*) and 7y and 7b share a great deal of sequence through recombination events (*Figure 4B–C*).

Next we used the newly generated RepeatMasker annotations to examine the RNA expression profiles of the different LTR7 subfamilies using scRNA-seq data from human preimplantation embryos and RNA-seq data from human ESCs (*Blakeley et al., 2015*; *Tang et al., 2010*) (see Materials and methods).

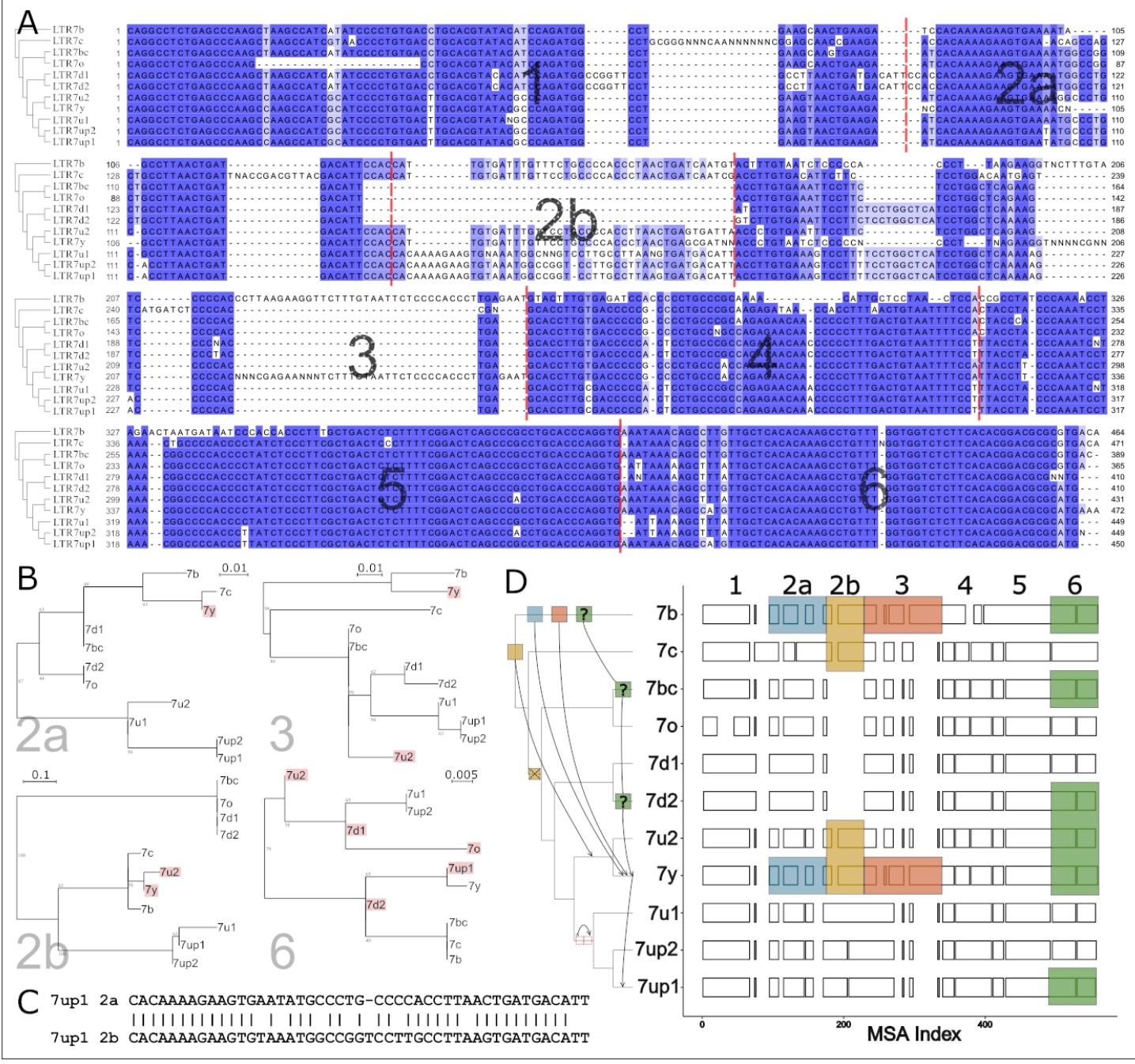

**Figure 4.** Modular block evolution of LTR7 subfamilies. (**A**) A multiple sequence alignment of LTR7 subfamily consensus sequences. The phylogenetic topology from *Figure 1* is shown on the left. The MSA is broken down into sequence blocks (red lines) with differential patterns of relationships. (**B**) Parsimony trees from **A** sequence blocks. Subfamilies whose blocks do not match the overall phylogeny are highlighted in red. Bootstrap values > 0 are shown. (**C**) Blastn alignment of LTR7up1 block 2a and 2b. (**D**) A multiple sequence alignment of majority rule consensus sequences from each LTR7 subfamily detailing shared structure. Blocks show aligned sequence; gaps represent absent sequence. Colored sections identify putative phylogeny-breaking events. Recombination events whose directionality can be inferred (via aging) are shown with blocks and arrows on the cladogram. Recombination events with multiple possible routes are denoted with '?'. The deletion of 2b is denoted on the cladogram with a red 'X'; the duplication of 2a is denoted with two red rectangles.

The online version of this article includes the following figure supplement(s) for figure 4:

**Figure supplement 1.** Parsimony tree of all consensus human endogenous retrovirus type-H (HERVH) long terminal repeats (LTR) blocks 2a and 2b.

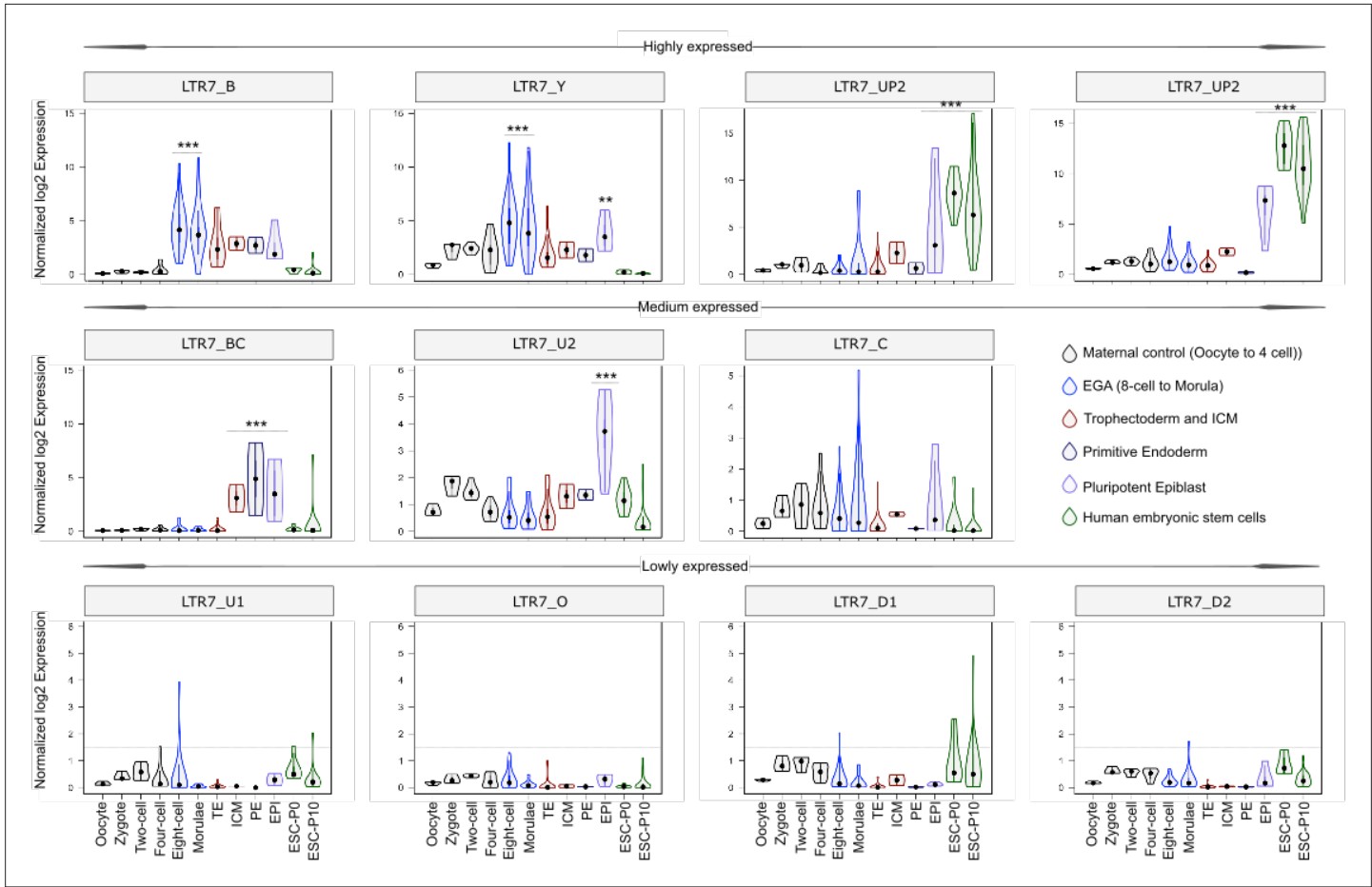

**Figure 5.** Expression profile of LTR7 subfamilies in human preimplantation embryonic lineages and embryonic stem cells (ESCs). The solid dots and lines encompassing the violins represent the median and quartiles of single cellular RNA expression. The color scheme is based on embryonic stages, defined as maternal control of early embryos (oocytes, zygote, 2 cell and 4 cell stage), EGA (8 cell and morula), inner cell mass (ICM), trophectoderm (TE), epiblast (EPI), and primitive endoderm (PE) from the blastocyst, and ESCs at passages 0 and 10.

The online version of this article includes the following figure supplement(s) for figure 5:

**Figure supplement 1.** Majority rule consensus sequences used for remasking of human genome.

**Figure supplement 2.** LTR7 subfamily expression in 'primed' and 'naïve' cell lines.

As expected, we found that the 7up subfamilies were highly expressed in the pluripotent epiblast and in ESCs (***Figure 5***). 7up expression was highly specific to these pluripotent cell types, with little to no transcription at earlier developmental timepoints. As previously observed (***Göke et al., 2015***), the 7b subfamily exhibited expression at the 8 cell and morula stages, coinciding with EGA (***Figure 5***). Another remarkable expression pattern was that of 7u2 which was restricted to the pluripotent epiblast (***Figure 5***). Interestingly, the 7y subfamily combined the expression of 7b and 7u2 (8 cell and morula plus epiblast), perhaps reflecting the acquisition of sequence blocks from both subfamilies (***Figure 4B–C***). Despite very similar sequence and age (***Figures 1, 2 and 4A***), 7bc and 7o elements show stark contrast in their expression profiles. 7o elements show no significant transcription at any timepoint in early development, while 7bc elements display RNA expression throughout the blastocyst, including trophectoderm and inner cell mass, primitive endoderm, and pluripotent epiblast (***Figure 5***). Previous expression analysis of the oldest LTR7 subfamily, 7c, did not find robust stage-specific expression (***Göke et al., 2015***). Our analysis revealed that some 7c elements display moderate RNA expression at various developmental stages (***Figure 5***). This pattern may reflect the relatively high level of sequence heterogeneity within this subfamily (***Figure 1***).

In summary, our analysis indicates that LTR7 subfamilies have distinct but partially overlapping expression profiles during human early embryonic development that appear to mirror their complex history of sequence diversification.

## A predicted SOX2/3 motif unique to 7up is required for transcriptional activity in pluripotent stem cells

We hypothesized that differences in embryonic transcription among LTR7 subfamilies were driven by the gain and loss of TF binding motifs, and that one or more of these mutations led to 7up's pluripotent-specific transcription. To find TF motifs enriched within each LTR7 subfamily relative to the others, we performed an unbiased motif enrichment analysis using the program HOMER to calculate enrichment scores of known TF motifs within each segmental block defined in *Figure 4A* in a pairwise comparison of each subfamily against each of the other subfamilies (see Materials and methods). The results yielded a slew of TF motifs enriched for each subfamily relative to the others (see *Figure 6A* for 7up1 and enrichment for all HERVH subfamilies in *Supplementary files 3 and 4*). These results suggested that each LTR7 subfamily possesses a unique repertoire of TF binding motifs, which could explain their differential expression during embryonic development.

Next, we sought to pinpoint mutational events responsible for the gain of TF motifs responsible for the unique expression of 7up in ESC. The single most striking motif distinguishing the 7up clade from the others was a SOX2/3 motif which coincided with an 8 bp insertion in block 2b (*Figure 6A, B*). Note this motif (and insertion) was also present in 7u1, the closest relative to 7up (*Figure 1*), but absent in all other subfamilies (*Figure 6B*).

We hypothesized that the 8 bp insertion provided a binding motif for SOX2 and/or SOX3 contributing to 7up promoter activity in ESCs. Indeed, SOX2 and SOX3 bind a highly similar motif (*Bergsland et al., 2011*; *Heinz et al., 2010*), activate an overlapping set of genes, and play a redundant function in pluripotency (*Corsinotti et al., 2017*; *Niwa et al., 2016*; *Wang et al., 2012*). In addition, we observed that both SOX2 and SOX3 are expressed in human ESCs but SOX3 was more specifically expressed in ESCs (*Figure 6—figure supplement 1A, C*). While SOX3 binding has not been profiled in human ESCs, ChIP-seq data available for SOX2 indicated that it binds preferentially 7up in a region coinciding with the 8 bp motif (*Figure 6B*). Together these observations suggest that 7up promoter activity in ESCs might be conferred in part by the gain of a SOX2/3 motif in block 2b.

To experimentally test this prediction, we used a luciferase reporter to assay promoter activity of three different LTR7 sequences in iPSCs (see Materials and methods). The first consisted of the full-length 7d consensus sequence (predicted to be inactive in iPSCs), the second contained the full-length 7up1 consensus (predicted to be active), and the third used the same 7up1 consensus sequence but lacking the 8 bp motif unique to 7up1/2 and 7u1 elements overlapping the SOX2/3 motif (*Figure 6B, C*). The results of the assays revealed that the 7d construct exhibited, as predicted, only weak promoter activity in iPSC compared to the empty vector (*Figure 6D*), while the 7up1 construct had much stronger promoter activity, driving on average 7.8-fold more luciferase expression than 7d and 100-fold more than the empty vector (*Figure 6D*). Strikingly, the promoter activity was essentially abolished in the 7up1 construct lacking the 8 bp motif, which drove minimal luciferase expression (on average, 3-fold less than LTR7d and 20-fold less than the intact LTR7up sequence). These results demonstrate that the 8 bp motif in 7up1 is necessary for robust promoter activity in iPSCs, likely by providing a SOX2/3 binding site essential for this activity.

## Discussion

The HERVH family has been the subject of intense investigation for its transcriptional and regulatory activities in human pluripotent stem cells. These studies often have treated the entire family as one homogenous, monophyletic entity and it has remained generally unclear which loci are transcribed and potentially important for pluripotency. This is in part because HERVH/LTR7 is an abundant and young family which poses technical challenges to interrogate the activity of individual loci and design experiments targeting specific members of the family (*Chuong et al., 2017*; *Lanciano and Cristofari, 2020*). Here, we applied a 'phyloregulatory' approach that integrates regulatory genomics data to a phylogenetic analysis of LTR7 sequences to reveal several new insights into the origin, evolution, and transcription of HERVH elements. In brief, our results show that: (i) LTR7 is a polyphyletic group

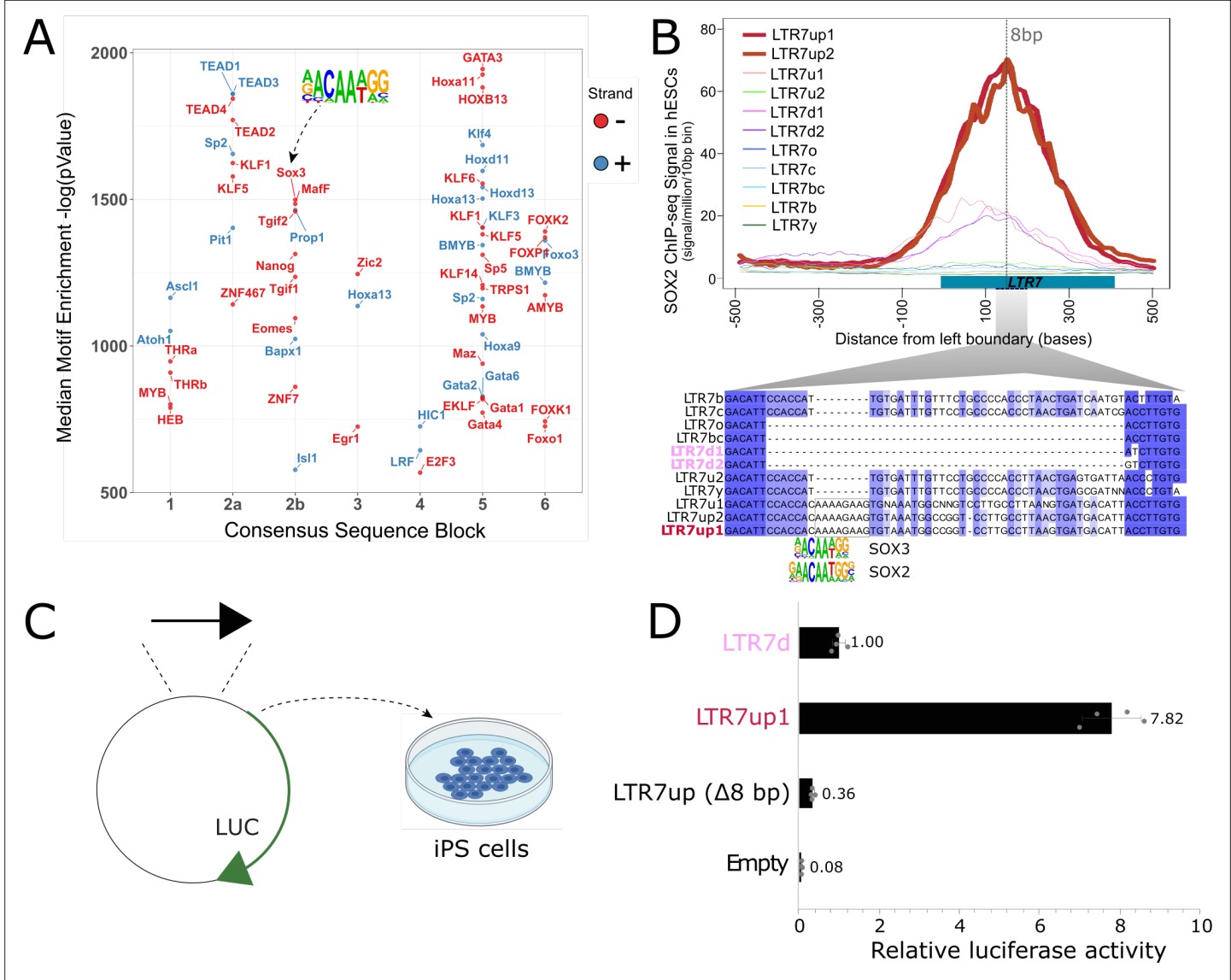

**Figure 6.** An 8 bp insertion, SOX2/3 binding site necessary for LTR7up transcription. (**A**) (log) p-values > 500 for HOMER motifs enriched in 7up1 insertion's sequence blocks vs. the same blocks from other insertions from other human endogenous retrovirus type-H (HERVH) subfamilies are shown. (**B**) Line plots show SOX2 ChIP-seq signal at LTR7 subfamily loci in human embryonic stem cells (ESCs). Signal from genomic loci was compiled relative to position 0. The 7up/u1 8 bp insertion position is shown with a dotted line. Region 2b harboring SOX2/3 transcription factor binding site (TFBS) is detailed below. (**C**) Scheme of DNA fragments cloned into pGL3-basic vector driving luciferase gene expression (LUC) with identified SOX2/3 motifs. Three constructs were analyzed: Entire LTR7up (7up1), 7d1/2 consensus sequence (approximate ancestral sequence for all LTR7d) and LTR7up with eight nucleotides deleted (LTR7up (Δ8bp – AAAAGAAG)) (see panel B). (**D**) Normalized relative luciferase activity of tested fragments compared to LTR7 down; n = 4 measurements; bars, means across replicates; error bars, standard deviation of the mean, dots, individual replicates.

The online version of this article includes the following figure supplement(s) for figure 6:

**Figure supplement 1.** Transcription factor (TF) expression in the preimplantation embryo and TF binding in embryonic stem cells.

**Figure supplement 2.** Heatmap showing read pileup from GRO-seq plus strand and SOX2 ChIP-seq on 7up and 7u1 insertions.

**Figure supplement 3.** Violin plots detailing the distribution of transcription factor (TF) expression shown in *Figure 6—figure supplement 1* in early human embryos and embryonic stem cells.

composed of at least eight monophyletic subfamilies; (ii) these subfamilies have distinct evolutionary histories and transcriptional profiles in human embryos and a single and relatively small subgroup (~264 loci), LTR7up, exhibits robust promoter activity in ESC; (iii) LTR7 evolution is characterized by the gain, loss, and exchange of cis-regulatory modules likely underlying their transcriptional partitioning during early embryonic development.

## Phyloregulatory analysis of LTR7 disentangles the cis-regulatory evolution of HERVH

Previous studies have treated LTR7 sensu stricto insertions as equivalent representatives of their subfamilies (*Bao et al., 2015*; *Gemmell et al., 2019*; *Göke et al., 2015*; *Izsvák et al., 2016*; *Storer et al., 2021*; *Wang et al., 2014*; *Zhang et al., 2019*). While some of these studies were able to detect differential transcriptional partitioning between LTR7, LTR7y, and LTR7b (*Göke et al., 2015*), the amalgamating of LTR7 loci limited the ability to detect transcriptional variations among LTR7 and to identify key sequence differences responsible for divergent transcription patterns. Our granular parsing of LTR7 elements and their phyloregulatory profiling has revealed striking genetic, regulatory, and evolutionary differences among these sequences. Importantly, a phylogeny based on the coding sequence (reverse transcriptase [RVT] domain) of HERVH provided less granularity to separate the subfamilies than the LTR sequences (*Figure 1—figure supplement 1*). The classification of new subfamilies within LTR7 enabled us to discover that they have distinct expression profiles during early embryonic development (*Figure 5*) that were previously obscured by their aggregation into a single group of elements. For example, the 7u2 subfamily is, to our knowledge, the first subfamily of human TEs reported to have preimplantation expression exclusively in the epiblast.

It has been observed for some time that only a small subset of HERVH elements are expressed in ESCs (*Gemmell et al., 2019*; *Göke et al., 2015*; *Ohnuki et al., 2014*; *Santoni et al., 2012*; *Schön et al., 2001*; *Wang et al., 2014*; *Zhang et al., 2019*). Some have attributed this property to variation in the internal region of HERVH, context-dependent effects (local chromatin or cis-regulatory environment) and/or age (*Gemmell et al., 2019*; *Zhang et al., 2019*). Our results provide an additional, perhaps simpler explanation: we found that HERVH elements expressed in ESCs are almost exclusively driven by two closely related subfamilies of LTR7 (7up) that emerged most recently in hominoid evolution. We identified one 8 bp sequence motif overlaps a predicted SOX2/3 binding site unique to the 7up lineage that is required for promoter activity in pluripotent stem cells. This exact motif is also present in the untranscribed 7u1 subfamily. More distantly related subfamilies also have SOX2/3 binding sites elsewhere on their sequence (*Supplementary file 4*), indicating that the presence of a SOX2/3 site is not sufficient to confer transcriptional activity in ESC. Additionally, the strength of the motif match does not correlate with SOX2 binding or transcription within 7up (*Figure 6—figure supplement 2*). Perhaps SOX2/3 cooperatively bind with other TF to uniquely activate 7up. These results highlight that the primary sequence of the LTR plays an important role in differentiating and diversifying HERVH expression during human embryonic development.

The phyloregulatory approach outlined in this study could be applied to illuminate the regulatory activities of LTR elements in other cellular contexts. In addition to embryogenesis, subsets of LTR7 and LTR7y elements are known to be upregulated in oncogenic states (*Babaian and Mager, 2016*; *Glinsky, 2015*; *Kong et al., 2019*; *Yu et al., 2013*). It would be interesting to explore whether these activities can be linked to the gain of specific TFBS using the new LTR7 annotations and regulatory information presented herein. Other human LTR families, such as MER41, LTR12C, or LTR13, have been previously identified as enriched for particular TF binding and cis-regulatory activities in specific cellular contexts (*Chuong et al., 2016*; *Deniz et al., 2020*; *Ito et al., 2017*; *Krönung et al., 2016*; *Sundaram et al., 2014*). In each case, TF binding enrichment was driven by a relatively small subset of loci within each family. We suspect that some of the intrafamilial differences in TF binding and cis-regulatory activity may be caused by unrecognized subfamily structure and subfamily-specific combinations of TFBS, much like we observe for LTR7.

## Recombination as a driver of LTR cis-regulatory evolution

Recombination is a common and important force in the evolution of exogenous RNA viruses (*Jetzt et al., 2000*; *Pérez-Losada et al., 2015*; *Simon-Loriere and Holmes, 2011*) and ERVs (*Vargiu et al., 2016*). Traditional models of recombination describe recombination occurring due to template

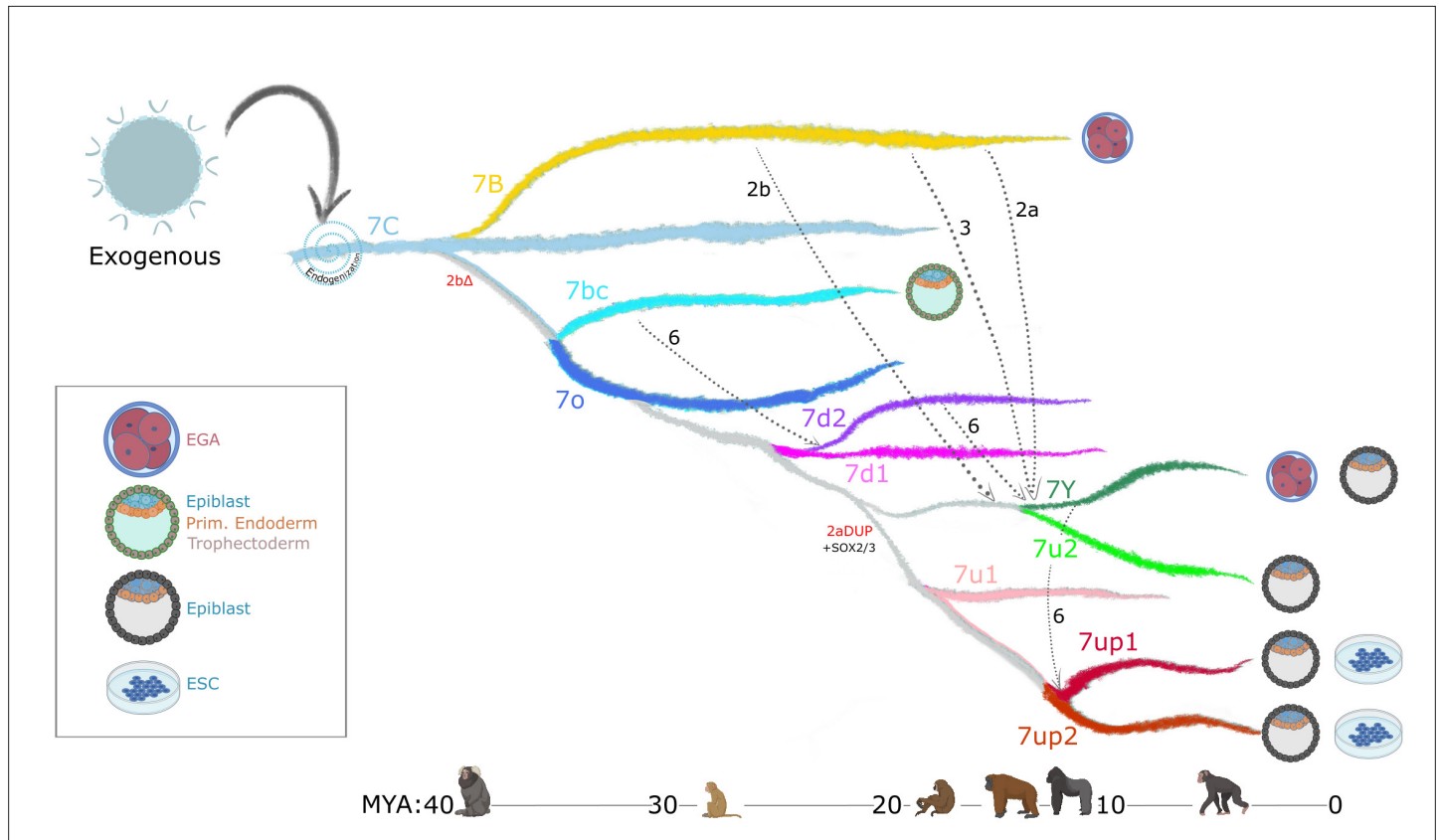

**Figure 7.** Model of LTR7 subfamily evolution. Estimated LTR7 subfamily transpositional activity in million years ago (mya) is listed with corresponding approximate primate divergence times (bottom). The positioning and duration of transpositional activity are based on analysis from *Figure 3B*. The gray connections between subfamilies indicate average tree topology which is driven by overall pairwise sequence similarity. Dashed lines indicate likely recombination events which led to the founding of new subfamilies. Stage-specific expression profiles from *Figure 5A* are detailed to the right of each corresponding branch.

switching during reverse transcription, a process that requires the co-packaging of RNA genomes, a feature of retroviruses and some retrotransposons (*Lai, 1992*; *Matsuda and Garfinkel, 2009*). Previous studies proposed that the HERVH family had undergone inter-element recombination events of both its coding genes (*Mager and Freeman, 1987*; *Vargiu et al., 2016*) and LTR (*Goodchild et al., 1993*). Specifically, it was inferred that recombination event between Type I LTR (i.e., LTR7) and Type II LTR (LTR7b) led to the emergence of Type Ia (LTR7y).

Our findings of extensive sequence block exchange between 7y and 7b (*Figure 4D*) are consistent with these inferences. Furthermore, our division of HERVH into at least 11 subfamilies, rather than the original trio (Type I, II, Ia), and systematic analysis of recombination events (*Figure 4*) suggest that recombination has occurred between multiple lineages of elements and has been a pervasive force underlying LTR diversification. We identified a minimum of six recombination events spanning 20 my of primate evolution (see *Figure 4D* and summary model in *Figure 7*). The coincidence of recombination events with changes in expression profiles (*Figure 7*) suggests that these events were instrumental to the diversification of HERVH embryonic expression. The hybrid origin and subsequent burst of amplification of LTR7 subfamilies (*Figures 1 and 2*) suggest they expanded rapidly after shifting their transcriptional profiles. The coincidence of niche colonization with a burst in transposition leads us to speculate that these shifts in expression were foundational to the formation and successful expansion of new HERVH subfamilies. It would be interesting to explore whether inter-element recombination has also contributed to the evolution of other LTR subfamilies and the diversification of their expression patterns.

Previous work has highlighted the role of TEs, and LTRs in particular, in donating built-in cis-regulatory sequences promoting the evolutionary rewiring of mammalian transcriptional networks

(*Chuong et al., 2017*; *Feschotte, 2008*; *Hermant and Torres-Padilla, 2021*; *Jacques et al., 2013*; *Rebollo et al., 2012*; *Sundaram and Wysocka, 2020*; *Thompson et al., 2016*). We show that recombination provides another layer to this idea, where combinations of TFBS can be mixed-and-matched, then mobilized and propagated, further accelerating the diversification of these regulatory DNA elements. As HERVH expanded and diversified, its newly evolved cis-regulatory modules became confined to specific host lineages (*Figure 2*). Thus, it is possible that the formation of new LTR via recombination and their subsequent amplification catalyzed cis-regulatory divergence across primate species.

## LTR evolution enabled HERVH's colonization of different niches in the human embryo

Our evolutionary analysis reveals that multiple HERVH subfamilies were transpositionally active in parallel during the past ~25 my of primate evolution (*Figures 2 and 7*). This is in stark contrast to the pattern of LINE1 evolution in primates, which is characterized by a single subfamily being predominantly active at any given time (*Khan et al., 2006*). We hypothesize that the ability of HERVH to colonize multiple cellular niches underlie this difference. Indeed, we observe that concurrently active HERVH subfamilies are transcribed at different developmental stages, such as 7up and 7u2 being transcribed in the pluripotent epiblast at the same time that 7y and the youngest 7b were transcribed at the 8 cell and morula stages (*Figure 7*). We posit that this partitioning allowed multiple HERVH subfamilies to amplify in parallel without causing overt genome instability and cell death during embryonic development.

Niche diversification may have also enabled HERVH to evade cell-type-specific repression by host-encoded factors such as KZFPs. KZFPs are thought to emerge and adapt during evolution to silence specific TE subfamilies in a cell-type-specific manner (*Bruno et al., 2019*; *Cosby et al., 2019*; *Ecco et al., 2017*; *Imbeault et al., 2017*). For example, there is evidence that the progenitors of the currently active L1HS subfamily became silenced in human ESCs via KZFP targeting, but evaded that repression and persisted in that niche through the deletion of the KZFP binding site (*Jacobs et al., 2014*). HERVH may have persisted through another evasive strategy: changing their TFBS repertoire to colonize niches lacking their repressors. To silence all LTR7, any potential HERVH-targeting KZFP would need to gain expression in multiple cellular contexts. For example, one potential repressor, ZNF534, binds a wide range of LTR7 sequences, but is particularly enriched at 7up in HEK293 cells (*Figure 3A, D*). Our analysis shows that ZNF534 is most highly expressed in the morula, but dips in human ESC (*Figure 3—figure supplement 2*). Thus, ZNF534 may repress 7up at earlier stages of development but is apparently unable to suppress 7up transcription in pluripotent stem cells. If true, this scenario would illustrate how LTR diversification facilitated HERVH persistence in the face of KZFP coevolution. Further investigation is needed to explore the interplay between KZFPs and HERVH subfamilies during primate evolution.

## Implications for stem cell and regenerative biology

Lastly, our findings may provide new opportunities for stem cell research and regenerative medicine. Our data on 7up reinforces previous findings (*Corsinotti et al., 2017*; *Wang et al., 2012*) that place SOX2/3 as central players in pluripotency. Furthermore, our analysis identified a set of TFs whose motifs are uniquely enriched in different LTR7 subfamilies with distinct expression patterns in early embryonic cells, which may enable a functional discriminatory analysis of the role of these TFs in each cell type. HERVH/LTR7 has been used as a marker for human pluripotency (*Ohnuki et al., 2014*; *Santoni et al., 2012*; *Wang et al., 2014*), and recent work has revealed that HERVH/LTR7-positive cells may be more amenable to differentiation, and are therefore referred to as 'primed' cells (*Göke et al., 2015*; *Theunissen et al., 2016*). However, primed cells are not as promising for regenerative medicine as so-called 'naïve' cells (*Nichols and Smith, 2009*), which are less differentiated and resemble cells from late morula to epiblast, or so-called 'formative' cells, which most closely resemble cells from the early post-implantation epiblast (*Kalkan and Smith, 2014*; *Kinoshita et al., 2021*; *Rossant and Tam, 2017*). Previous pan-LTR7 knockdown experiments have focused on highly transcribed LTR7 in primed ESCs (*Lu et al., 2014*; *Ohnuki et al., 2014*; *Wang et al., 2012*). As such, these experiments have primarily, if not exclusively, targeted 7up elements. While the role of individual 7up copies in the maintenance of pluripotency of primed cells remains to be clarified (*Takahashi et al.,*

*2021*), our work highlights potential additional roles of other LTR7 subfamilies in naïve, formative ESCs. Of relevance to this issue is our finding that elements of the 7u2 subfamily are highly and exclusively expressed in the pluripotent epiblast in vivo (*Figure 5*), but weakly so in H1 ESC, which consists of a majority of primed cells and a minority of naïve or formative cells (*Gafni et al., 2013*). Indeed, while 7up shows expression preference in primed cell lines, 7u2 shows preference for naïve lines (*Figure 5—figure supplement 2*). Thus, it might be possible to develop an LTR7u2-driven reporter system to mark and purify naïve or formative cells from an heterogenous ESC population. Similarly, a MERVL LTR-GFP transgene has been used in mouse to purify rare 2-cell-like totipotent cells where this LTR is specifically expressed amidst mouse ESCs in culture (*Hermant and Torres-Padilla, 2021*; *Macfarlan et al., 2012*).

In conclusion, our study highlights the modular cis-regulatory evolution of an ERV which has facilitated its transcriptional partitioning in early embryogenesis. We believe that phyloregulatory dissection of endogenous retroviral LTRs has the potential to further our understanding of the evolution, impact, and applications of these elements in a broad range of biomedical areas.

## Materials and methods

### HERVH LTR sequence identification

All HERVH-int and accompanying LTRs (LTR7, 7b, 7c, and 7y) were extracted from masked (Repeat-Masker version 4.0.5 repeat Library 20140131 – *Smit et al., 2013*) GRCh38/hg38 (alt chromosomes removed). All annotated HERVH-int and HERVH LTR were run through OneCodeToFindThemAll.pl (*Bailly-Bechet et al., 2014*) followed by rename_mergedLTRelements.pl (*Thomas et al., 2018*) to identify solo and full-length HERVH insertions. 5′ LTRs from full-length insertions >4 kb were combined with and solo LTRs. LTRs > 350 bp were considered for future analysis.

### Multiple sequence alignment, phylogenetic tree generation, and LTR7 subdivision

All HERVH LTRs (*Figure 1A – Supplementary file 5*) or only LTR7s (*Figure 1B – Supplementary file 6*) were aligned with mafft -auto (*Nakamura et al., 2018*) strategy: FFT-NS-2/Progressive method followed by PRANK (*Löytynoja and Goldman, 2010*) with options -showanc -support -njtree -uselogs -prunetree -prunedata -F -showevents. Uninformative structural variations were removed with Trimal (*Capella-Gutiérrez et al., 2009*) with option -gt 0.01.

To visualize inter-insertion relationships, the MSA was input into IQtree with options -nt AUTO -m MFP -bb 6000 -asr -minsup.95 (*Chernomor et al., 2016*). This only displays nodes with UFbootstrap support >0.95.

Clusters of >10 insertions sharing a node with UFbootstrap support that were separated from other insertions by internal branch lengths >0.015 (1.5 subs/100 bp) were defined as belonging to a new bona fide LTR7 subfamily (*Figure 1B*).

### LTR7 consensus generation and network analysis

Majority rule (51%) was used to generate each LTR7 subfamily at nodes described in *Figure 1*. Positions without majority consensus are listed as 'N'. Majority rule consensus sequences were aligned with MUSCLE in SEAVIEW (*Supplementary file 7*; *Edgar, 2004*; *Gouy et al., 2010*). Alignment was visualized with Jalview2 (*Waterhouse et al., 2009*; *Figure 4A*) and ggplot2 (*Figure 4*).

Non-gap SNPs from the muscle alignment were used to construct a median-joining network (*Bandelt et al., 1999*) with POPART (*Leigh and Bryant, 2015*).

### RVT domain extraction, alignment, and tree generation

The RVT domain was extracted from HERVH-int consensus via repbrowser (*Fernandes et al., 2020*):

CACCCTTACCCCGCTCAATGCCAATATCCCATCCCACAGCATGCTTTAAAAGGATTAAAGCCTG
TTATCACTCGCCTGCTACAGCATGGCCTTTTAAAGCCTATAAACTCTCCTTACAATTCCCCCATTTTA
CCTGTCCTAAAACCAGACAAGCCTTACAAGTTAGTTCAGGATCTGTGCCTTATCAACCAAATTG
TTTTGCCTATCCACCCCATGGTGCCAAACCCATATACTCTCCTATCCTCAATACCTCCCTCCACAACC
CATTATTCTGTTCTGGATCTCAAACATGCTTTCTTTACTATTCCTTTGCACCCTTCATCCCAGCCTCT
CTTCGCTTTCACTTGGA.

This sequence was blated (best hit) against all annotated HERVH-int in the human genome and matches were extracted. Corresponding LTR7 subdivision annotations from *Figure 1* were matched with these HERVH-int RT domains. Mafft alignment and IQTree generation were done identically to the Mafft and IQTree run for the LTRs (see corresponding Materials and methods section).

### Peak calling

ChIP-seq datasets representing TFs, histone modifications, and regulatory complexes in human ESCs and differentiated cells were retrieved from GSE61475 (38 distinct TFs and histone modifications), GSE69647 (H3K27Ac, POU5F1, MED1, and CTCF), GSE117395 (H3K27Ac, H3K9Me3, KLF4, and KLF17), and GSE78099 (an array of KRAB-ZNFs and TRIM28) (*Imbeault et al., 2017*). ZNFs enriched in LTR7 binding (ZNF90, ZNF534, ZNF75, ZNF69B, ZNF257, ZNF57, and ZNF101) from HEK293 peaks were all evaluated, but only ZNF90 and ZNF534 bound >100 LTR7 insertions (data not shown). The others were dropped from the analysis.

ChIP-seq reads were aligned to the hg19 human reference genome using the Bowtie2. All reads with phred score <33 and PCR duplicates were removed using bowtie2 and Picard tools, respectively. ChIP-seq peaks were called by MACS2 with the parameters in 'narrow' mode for TFs and 'broad' mode for histone modifications, keeping false discovery rate (FDR) < 1%. ENCODE-defined black-listed regions were excluded from called peaks. For phyloregulatory analysis (*Figure 2*), we then converted hg19 to hg38 (no alt) coordinates via UCSC *liftover* (100% of coordinates lifted) and inter-sected these peak with the loci from LTR7 subfamilies using bedtools with any overlap. For ChIP-seq binding enrichment on a subset of marks following motif analysis (*Figure 5*), 70% overlap of peak and LTR7 was required. Enrichment of a given TF within LTR7 subfamilies was calculated using enrichR package in R, using the customized in-house codes (see the codes on GitHub for the detailed analysis pipelines and calculation of enrichment score).

### Phyloregulatory analysis

Peaks from external ChIP-seq datasets were intersected with LTR7 insertions (*Quinlan and Hall, 2010*). LTR7 insertions that intersected with >1 bp of peaks were counted as positive for the respective mark. We repeated this analysis with a range of overlap requirements from extending the LTR 500 bp into unique DNA to 70% overlap and found few differential calls (data not shown). The phylogenetic tree rooted on 7b (ggtree) was combined with these binary data (ggheat).

'Highly transcribed' (fpkm >2) and 'chimeric' HERVH from H1 cells (GSE54726) (*Wang et al., 2014*) were intersected with LTR7 similarly to ChIP-seq data. Those which intersected LTR7 were marked as 'RNA-seq' or 'chimeric', respectively. GRO-seq profiles from H1 cells (*Estarás et al., 2015*) (GSE64758) were created for windows 10 bp upstream and 8 kb downstream of 5' and solo LTR7 (*Ramírez et al., 2016*). The most visible signal was confined to the top 7th of insertions (*Figure 3—figure supplement 1*). All LTR7 were subdivided into septiles, due to visible signal being confined to the top 7th of inser-tions; those of the top septile were labeled 'GRO-seq'.

### Peak proportion heatmap generation and statistical analysis

Tables with the proportion of solo and 5' LTRs from a given subfamily positive for select marks (phylo-regulatory analysis) were used to generate heatmaps with the R package ggplot (ggheat) (*Ginestet, 2011*). Those with padj < 0.05 (chi-square Bonferroni correction n = 147 tests for a total of 21 marks examined) were considered significantly enriched in 7up1. Enrichment for non-LTR7up subfamilies was not tested. While not all tested marks are displayed in the main text, statistical analysis was performed with all tested marks (n = 147) (*Supplementary file 8*). For comparing transcribed 7up to untran-scribed 7up, 18 pairwise comparisons were made (*Supplementary file 9*).

### Aggregate signal heatmap generation

GRO-seq (H1 cells – GSE64758), whole-genome bisulfite sequencing (WGBS-seq – H1 cells), SOX2 GEO ID GSE125553 (*Bayerl et al., 2021*), and H3K9me3 ChIP-seq (H1 – primed – GSE78099) bams were retrieved from *Estarás et al., 2015*, *ENCODE Project Consortium, 2012*, and *Theunissen et al., 2016*, respectively. Deeptools (*Ramírez et al., 2016*) was used to visualize these marks by LTR7 subfamily division in windows 500 bp upstream and 8 kb downstream of the most 5' position in the LTR (*Figure 3—figure supplement 1*, *Figure 6—figure supplement 2*).

### Orthologous insertion aging

Human coordinates for 7b, 7c, and 7y and LTR7 used in alignments and tree generation were lifted over (*Kent et al., 2002*; *Raney et al., 2014*) from GRCh38/hg38 (*Miga et al., 2014*) to Clint_PTRv2/panTro6 (*Waterson et al., 2005*), Kamilah_GGO_v0/gorGor6 (*Scally et al., 2012*), Susie_PABv2/ponAbe3 (*Locke et al., 2011*), GGSC Nleu3.0/nomLeu3 (*Carbone et al., 2014*), or Mmul_10/rheMac10 (*Gibbs et al., 2007*). Those that were successfully lifted over from human to non-human primates (syntenic regions) were then lifted over back to human. Only those that survived both lift-overs (1:1 orthologous) were counted as present in non-human primates. The proportion of those orthologous to human and total number of orthologous was plotted with ggplot2.

### Terminal branch length aging

Terminal branch lengths from the LTR7 phylogenetic tree (*Figure 1B*) were extracted and plotted with ggplot2. Similarly aged subfamilies were inferred from means here and from orthologous insertion aging for statistical testing. Three total groups were tested for differences in means (7up1/7up2/7u2 vs. 7d1/7d2/7u1 vs. 7bc/o) via Wilcoxon rank-sum test with Bonferroni multiple testing correction.

### Identification of recombination breakpoints and consensus parsimony tree generation

Major recombination breakpoints were identified by eye from the consensus sequence MSA, where SNPs and structural rearrangements seemed to have different relationships between blocks. Putative block recombination events were identified by looking for shared shapes in the block consensus MSA (*Figure 4A*). To test if these were truly recombination events and could not be explained by evolution by common descent, inter-block sequence relationship differences were tested by generating parsimony trees and comparing these to the overall phylogenetic structure from *Figure 1A*. Parsimony trees were generated in SEAVIEW, treating all gaps as unknown states (except in the case of 2b, where the entire sequence is gaps and gaps were not treated differently than other sequence), bootstrapped 5000 times with the option 'more thorough tree search'. Differences in block parsimony trees and the overall phylogeny that had bootstrap support were marked in red and included in *Figures 4D and 7*.

### 7up consensus block 2a and 2b alignment and parsimony tree

LTR7up blocks 2a and 2b (*Figure 4*) appeared to share sequence. To determine if block 2b was the result of a duplication of 2a, we extracted these sequences from the LTR7up1 consensus and aligned them with blastn (NCBI web version) with default settings. To determine the relationship of all HERVH LTR 2a and 2b blocks, we performed a muscle alignment (default settings) of all 2a and 2b from all HERVH LTR consensus sequences and then generated a parsimony tree with 5000 bootstraps with SEAVIEW with the option 'more thorough tree search'.

### New LTR7B/C/Y consensus generation and remasking of human genome

Consensus sequences for LTR7 subfamilies were generated using the tree from *Figure 1b* (see above). For LTR7b/c/y, we used the alignment and tree comprising all HERVH LTR (*Figure 5—figure supplement 1*). To do this, we identified nodes with >0.95 UFbootstrap support that were comprised of predominately (>90%) of previously annotated LTR7b, LTR7c, or LTR7y. These sequences were used to generate majority rule consensus sequences for their respective subfamily. We generated two mutually exclusive LTR7c consensus sequences (LTR7C1 and LTR7C2) due to the high sequence divergence of LTR7C. Both of these subfamilies were merged into 'LTR7C' after remasking.

Parsing previously annotated LTR7 into eight subfamilies and evidence of recurrent recombination events caused concern that HERVH LTRs may be misannotated in the RepeatMasker annotations. To compensate, we remasked (*Smit et al., 2013*) GRCh38/hg38 (excluding alt chromosomes) with a custom library consisting of the new consensus sequences for LTR7 subfamilies, new consensus sequences for 7b, 7y, and 7c (see above) based on the HERVH LTR tree from *Figure 4*, and HERVH-int (dfam). We also included annotated consensus sequences from dfam for MER48, MER39, AluYk3, and MST1N2, who we found an HERVH-only library also masked to a limited degree (data not shown). With this library, we ran RepeatMasker with crossmatch and 'sensitive' settings: -e crossmatch -a -s -no_is. Changes in annotations can be found in (HERVH_LTRremasking.xlsx).

## Embryonic HERVH subfamily expression analysis

We downloaded the raw single-cell RNA-seq datasets from early human embryos and ESCs (GSE36552) and the EPI, PE, TE cells (GSE66507) in sra format. Following the conversion of raw files into fastq format, the quality was determined by using the FastQC. We removed two nucleotides from the ends as their quality scores were highly variable compared with the rest of the sequences in RNA-seq reads. Prior to aligning the resulting reads, we first curated the reference genome annotations using the LTR7 classification, as shown in the manuscript. We extracted the genes (genecode V19) and LTR7 subfamilies (see *Figure 5*) genomic sequences and combined them to generate a reference transcriptome. These sequences were then appended, comprising the coding sequences plus UTRs of genes and locus-level LTR7 subfamilies sequences in fasta format. We then annotated every fasta sequences with their respective genes or LTR7 subfamilies IDs. To guide the transcriptome assembly, we also appended the each of the resulting contigs and modelled them in gtf format that we utilized for the expression quantification. Next, we indexed the concatenated genes and LTR7 subfamilies transcriptome and genome reference sequences using 'salmon' (*Patro et al., 2017*). Finally, we aligned the trimmed sequencing reads against the curated reference genome. The 'salmon' tool quantified the counts and normalized expression (transcripts per million [TPM]) for each single-cell RNA-seq sample. Overall, this approach enabled us to simultaneously calculate LTR7 subfamilies and protein-coding gene expression using expected maximization algorithms. Data integration of obtained count matrix, normalization at logarithmic scale, and scaling were performed as per the 'Seurat V.3.7' (http://sati-jalab.org/seurat/) guidelines. The annotations of cell types were taken as it was classified in original studies. We calculated differential expression and tested their significance level using Kruskal-Wallis test by comparing cell types of interest with the rest of the cells. The obtained p-values were further adjusted by the Benjamini-Hochberg method to calculate the FDR. All the statistics and visualization of RNA-seq were performed on R (https://www.r-project.org/).

## Motif enrichment

For each subfamily of LTR7 elements, all re-annotated elements were aligned against the subfamily consensus sequence using MUSCLE (*Edgar, 2004*). These multiple sequence alignments were then split based on the recombination block positions in the consensus sequence. The consensus sequence was then removed. Binding motif position-weight matrices were downloaded from HOMER (*Heinz et al., 2010*) and were used to perform pairwise motif enrichment using the command 'homer2 find'. For LTR7up1 enrichment (*Figure 6A* – testing which motifs were enriched in LTR7up1 compared to other subfamilies), enrichment was only calculated for LTR7up1 and the motifs with a -log(p-value) cutoff of $1 \times 10^{-5}$ were kept. For enrichment in all subfamilies (*Supplementary files 3 and 4*) – testing all subfamilies against all others, every pairwise subfamily combination within each block was tested and all results are displayed.

## SOX2 ChIP-seq signal on LTR7

SOX2 ChIP-seq and whole-cell extract datasets from primed human ESCs were downloaded in fastq format from GEO ID GSE125553 (*Bayerl et al., 2021*). Fastq reads were mapped against the hg19 reference genome with the bowtie2 parameters: -*very-sensitive-local*. All unmapped reads with Phred score <33 and putative PCR duplicates were removed using *Picard* and *samtools*. This analysis was repeated with exclusively uniquely mapping reads which did not affect the outcome of analysis (data not shown). All the ChIP-seq narrow peaks were called by MACS2 (FDR < 0.01). To generate a set of unique peaks, we merged ChIP-seq peaks within 50 bp of one another using the *mergeBed* function from bedtools. We then intersected these peak sets with LTR7 subgroups from hg19 repeat-masked coordinates using bedtools *intersectBed* with 50% overlap. LTR7up1 and LTR7up2 were harboring the highest number of peaks compared with the rest of the subgroups. To illustrate the enrichment over the LTR7 subgroups, we first extended 500 basepairs from upstream and downstream coordinates from the left boundary of each LTR7subgroups. These 1 kb windows were further divided into 10 bps bins. The normalized ChIP-seq signal over the local lambda (piled up bedGraph outputs from MACS2) was counted in each bin. These counts were then normalized by the total number of mappable reads per million in given samples and presented as signal per million per 10 bps. Finally, these values were averaged across the loci for each bin to illustrate the subfamilies' level of ChIP-seq enrichment.

Replicates were merged prior to plotting. Note: Pearson's correlation coefficient between replicates across the bins was found to be $r > 0.90$.

### SOX3 motif strength correlation with SOX2 binding

All 7up1, 7up2, and 7u1 loci used in other analyses were ranked based on the strength/presence of the SOX3 motif in block 2b. GRO-seq and SOX2 (see above data sources) reads were layered onto these loci in windows 500 bp upstream and 8 kb downstream.

### Luciferase reporter assay

The inserts (LTR7 variants or EF1a promoter) with restriction enzyme overhangs were ordered from Genewiz and cloned into pGL3-basic plasmid upstream of the firefly reporter gene (Promega, #E1751). Minipreps were prepared with QIAprep Spin Miniprep kit (Qiagen, #27104). Plasmids were sequenced to ensure the correct sequence and directionality of the insert. Twenty-four hours before transfection, human iPSC WTC-11 (Coriell Institute, GM25256) cells were plated on Vitronectin (Thermo Fisher Scientific, #A14700) coated 12-well plates in Essential 8 Flex medium (Thermo Fisher Scientific, #A2858501) with E8 supplement, Rock inhibitor (STEMCELL Technologies, #72304) and 2.5% penicillin-streptomycin. Cells were co-transfected with 800 ng of plasmid of interest and 150 ng plasmid containing EF1a upstream of GFP for normalization with Lipofectamine Stem transfection reagent (Thermo Fisher Scientific, #STEM00008) according to manufacturer's instructions. Forty-eight hours after transfection, cell pellet was harvested and luciferase activity was measured with Luciferase Reporter Assay kit (Promega, #E1910) on Glomax (Promega) according to instructions. Transfection efficiency and cell count was normalized with GFP.

1.7down:
GCTAGCTGTCAGGCCTCTGAGCCCAAGCTAAGCCATCATATCCCCTGTGACCTGCACGTA
CACATCCAGATGGCCGGTTCCTGCCTTAACTGATGACATTCCACCACAAAAGAAGTGAAA
ATGGCCTGTTCCTGCCTTAACTGATGACATTATCTTGTGAAATTCCTTCTCCTGGCTCATCCTG
GCTCAAAAGCTCCCCTACTGAGCACCTTGTGACCCCCACTCCTGCCCGCCAGAGAACAAC
CCCCCTTTGACTGTAATTTTCCTTTACCTACCCAAATCCTATAAAACGGCCCCACCCCTATCTC
CCTTCGCTGACTCTCTTTTCGGACTCAGCCCGCCTGCACCCAGGTGAAATAAACAGCTTT
ATTGCTCACACAAAGCCTGTTTGGTGGTCTCTTCACACGGACGCGCATGCTCGAG
2.LTR7upcons:
GCTAGCTGTCAGGCCTCTGAGCCCAAGCCAAGCCATCGCATCCCCTGTGACTTGCACGTA
TACGCCCAGATGGCCTGAAGTAACTGAAGAATCACAAAAGAAGTGAATATGCCCTGCCCC
ACCTTAACTGATGACATTCCACCACAAAAGAAGTGTAAATGGCCGGTCCTTGCCTTAAGT
GATGACATTACCTTGTGAAAGTCCTTTTCCTGGCTCATCCTGGCTCAAAAAGCACCCCCA
CTGAGCACCTTGCGACCCCCACTCCTGCCCGCCAGAGAACAAACCCCCTTTGACTGTAAT
TTTCCTTTACCTACCCAAATCCTATAAAACGGCCCCACCCTTATCTCCCTTCGCTGACTCTCTT
TTCGGACTCAGCCCGCCTGCACCCAGGTGAAATAAACAGCCATGTTGCTCACACAAAGCC
TGTTTGGTGGTCTCTTCACACGGACGCGCATGCTCGAG
5.LTR7upcons_AAAGAAG_deletion:
GCTAGCTGTCAGGCCTCTGAGCCCAAGCCAAGCCATCGCATCCCCTGTGACTTGCACGTA
TACGCCCAGATGGCCTGAAGTAACTGAAGAATCACAAAAGAAGTGAATATGCCCTGCCCC
ACCTTAACTGATGACATTCCACCATTGTAAATGGCCGGTCCTTGCCTTAAGTGATGACAT
TACCTTGTGAAAGTCCTTTTCCTGGCTCATCCTGGCTCAAAAAGCACCCCCACTGAGCAC
CTTGCGACCCCCACTCCTGCCCGCCAGAGAACAAACCCCCTTTGACTGTAATTTTCCTTT
ACCTACCCAAATCCTATAAAACGGCCCCACCCTTATCTCCCTTCGCTGACTCTCTTTTCG
GACTCAGCCCGCCTGCACCCAGGTGAAATAAACAGCCATGTTGCTCACACAAAGCCTGTT
TGGTGGTCTCTTCACACGGACGCGCATGCTCGAG
5'NheI highlighted in yellow
3'XhoI highlighted in cyan

## Acknowledgements

We would like to thank Arian Smit and Robert Hubley for their assistance in remasking the human genome with the custom LTR7 library in accordance with RepeatMasker/Dfam specifications. We would also like to thank the Cornell Statistical Consulting Unit (CSCU), and Stephen Parry in particular, for their help with statistical analyses. Finally, we would like to credit BioRender.com for its primate

(*Figures 2 and 7*) and embryonic cell (*Figures 6 and 7*) graphics. This work was supported by funds from the National Institutes of Health to CF (GM112972; HG009391; GM122550); Cornell Center for Vertebrate Genomics to TAC; and the Cornell Presidential Fellow Program to MS. JLR is supported by the Howard Hughes Medical Institute Faculty Scholar program and NIH (GM099117).

## Additional information

### Funding

| Funder | Grant reference number | Author |
| --- | --- | --- |
| National Institutes of Health | GM112972 | Cédric Feschotte |
| National Institutes of Health | HG009391 | Cédric Feschotte |
| National Institutes of Health | GM122550 | Cédric Feschotte |
| Cornell Center for Vertebrate Genomics | | Thomas A Carter |
| Howard Hughes Medical Institute | | John L Rinn |
| National Institutes of Health | GM099117 | John L Rinn |
| Cornell Presidential Fellow Program | | Manvendra Singh |

The funders had no role in study design, data collection and interpretation, or the decision to submit the work for publication.

### Author contributions

Thomas A Carter, Conceptualization, Data curation, Formal analysis, Investigation, Methodology, Validation, Visualization, Writing – original draft, Writing – review and editing; Manvendra Singh, Data curation, Formal analysis, Investigation, Methodology, Validation, Visualization, Writing – review and editing; Gabrijela Dumbović, Jason D Chobirko, Formal analysis, Investigation, Methodology, Validation, Visualization, Writing – review and editing; John L Rinn, Cédric Feschotte, Conceptualization, Funding acquisition, Methodology, Project administration, Supervision, Writing – review and editing

### Author ORCIDs

Thomas A Carter http://orcid.org/0000-0001-7081-3259
Manvendra Singh http://orcid.org/0000-0002-8626-5418
Jason D Chobirko http://orcid.org/0000-0001-8495-9152
John L Rinn http://orcid.org/0000-0002-7231-7539
Cédric Feschotte http://orcid.org/0000-0002-8772-6976

### Decision letter and Author response

Decision letter https://doi.org/10.7554/eLife.76257.sa1
Author response https://doi.org/10.7554/eLife.76257.sa2

## Additional files

### Supplementary files

• Supplementary file 1. List of all LTR7 insertions included in phyloregulatory analysis in hg38 coordinates with accompanying binary regulatory calls.

• Supplementary file 2. OneCodeToFindThemAll.pl calls for remasked LTR7, LTR7B, LTR7C, and LTR7Y (hg38 coordinates) and quantification of subfamily annotation changes between old and new repeat masking.

- Supplementary file 3. Complete statistical support for HOMER motif enrichment for LTR7 subfamilies by sequence block.
- Supplementary file 4. Most enriched HOMER motifs for each human endogenous retrovirus type-H (HERVH) subfamily by sequence block.
- Supplementary file 5. Alignment of all human endogenous retrovirus type-H (HERVH) long terminal repeats (LTRs) used in study (including 7b/c/y).
- Supplementary file 6. Alignment of only LTR7 long terminal repeats (LTRs) used in study.
- Supplementary file 7. Alignment of human endogenous retrovirus type-H (HERVH) long terminal repeats (LTR) consensus sequences.
- Supplementary file 8. Full statistical support for phyloregulatory analysis.
- Supplementary file 9. Statistical support for transcribed vs. non-transcribed LTR7up1/2.
- Transparent reporting form

### Data availability

Scripts, data tables, and notes for figures 1-4,6a and figure supplements 1-1,2-1,3-1,4-1,5-1,6-2 - https://github.com/LumpLord/Mosaic-cis-regulatory-evolution-drives-transcriptional-partitioning-of-HERVH-endogenous-retrovirus../ copy archived at swh:1:rev:c5e3c56fdeb74511786be120d262393f18fea185. Scripts and data tables by MS for figures 5,6c and figure supplements 6-1,6-3,5-2 - https://github.com/Manu-1512/LTR7-up copy archived at swh:1:rev:23b4f17bc5c40f2992dcca264ed3b14fed84555b.

The following previously published datasets were used:

| Author(s) | Year | Dataset title | Dataset URL | Database and Identifier |
|---|---|---|---|---|
| Tsankov AM, Gu H, Akopian V, Ziller MJ, Donaghey J, Amit I, Gnirke A, Meissner A | 2015 | Transcription factor binding dynamics during human ES cell differentiation | https://www.ncbi.nlm.nih.gov/geo/query/acc.cgi?acc=GSE61475 | NCBI Gene Expression Omnibus, GSE61475 |
| Ji X, Dadon DB, Powell BE, Fan ZP, Borges-Rivera D, Shachar S, Weintraub AS, Hnisz D, Pegoraro G, Lee TI, Misteli T, Jaenisch R, Young RA | 2015 | 3D Chromosome Regulatory Landscape of Human Pluripotent Cells | https://www.ncbi.nlm.nih.gov/geo/query/acc.cgi?acc=GSE69647 | NCBI Gene Expression Omnibus, GSE69647 |
| Pontis J, Planet E, Offner S, Turelli P, Duc J, Coudray A, Theunissen TW, Jaenisch R, Trono D | 2019 | Hominid-specific transposable elements and KRAB-ZFPs facilitate human embryonic genome activation and transcription in naïve hESCs | https://www.ncbi.nlm.nih.gov/geo/query/acc.cgi?acc=GSE117395 | NCBI Gene Expression Omnibus, GSE117395 |
| Imbeault M, Helleboid PY, Trono D | 2017 | ChIP-exo of human KRAB-ZNFs transduced in HEK 293T cells and KAP1 in hES H1 cells | https://www.ncbi.nlm.nih.gov/geo/query/acc.cgi?acc=GSE78099 | NCBI Gene Expression Omnibus, GSE78099 |
| Wang J, Xie G, Singh M, Ghanbarian AT, Raskó T, Szvetnik A, Cai H, Besser D, Prigione A, Fuchs NV, Schumann GG, Chen W, Lorincz MC, Ivics Z, Hurst LD, Izsvák Z | 2014 | Repeat elements study in pluripotent stem cells | https://www.ncbi.nlm.nih.gov/geo/query/acc.cgi?acc=GSE54726 | NCBI Gene Expression Omnibus, GSE54726 |
| Estarás C, Benner C, Jones KA | 2015 | Wnt3a-Activin A Synergy Induces eRNAPII Pause-Release and Counteracts a Yap1 Elongation Block during hESC Differentiation | https://www.ncbi.nlm.nih.gov/geo/query/acc.cgi?acc=GSE64758 | NCBI Gene Expression Omnibus, GSE64758 |

*Continued on next page*

*Continued*

| Author(s) | Year | Dataset title | Dataset URL | Database and Identifier |
|---|---|---|---|---|
| Bayerl J, Ayyash M, Shani T, Manor YS, Gafni O, Massarwa R, Kalma Y, Aguilera-Castrejon A, Zerbib M, Amir H, Sheban D, Geula S, Mor N, Weinberger L, Naveh Tassa S, Krupalnik V, Oldak B, Livnat N, Tarazi S, Tawil S, Wildschutz E, Ashouokhi S, Lasman L, Rotter V, Hanna S, Ben-Yosef D, Novershtern N, Viukov S, Hanna JH | 2021 | Principles of Signalling Pathway Modulation for Enhancing Human Naïve Pluripotency Induction [ChIP-seq] | https://www.ncbi.nlm.nih.gov/geo/query/acc.cgi?acc=GSE125553 | NCBI Gene Expression Omnibus, GSE125553 |
| Yan L, Yang M, Guo H, Yang L, Wu J, Li R, Liu P, Lian Y, Zheng X, Yan J, Huang J, Li M, Wu X, Wen L, Lao K, Li R, Qiao J, Tang F | 2013 | Tracing pluripotency of human early embryos and embryonic stem cells by single cell RNA-seq | https://www.ncbi.nlm.nih.gov/geo/query/acc.cgi?acc=GSE36552 | NCBI Gene Expression Omnibus, GSE36552 |
| Gerri C, McCarthy A, Alanis-Lobato G, Demtschenko A, Bruneau A, Loubersac S, Fogarty NME, Hampshire D, Elder K, Snell P, Christie L, David L, Van de Velde H, Fouladi-Nashta AA, Niakan KK | 2015 | Single-Cell RNA-seq Defines the Three Cell Lineages of the Human Blastocyst | https://www.ncbi.nlm.nih.gov/geo/query/acc.cgi?acc=GSE66507 | NCBI Gene Expression Omnibus, GSE66507 |

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
