## [Editor Report]

Transposons achieve evolutionary success only upon self-replication in the germline, where novel genomic insertions are passed to the next generation. To define the genetic determinants of transposon specialization to the germline and specifically, specialization to pluripotent embryonic stem cells, this article applies evolutionary analysis, cell type-specific transcriptomics, and expression reporter assays to the human endogenous retrovirus type-H, HERV. Previous links between HERV and the maintenance of pluripotency make the discoveries consequential for the mobile element and genome evolution communities along with those engaged in stem cell biology and regenerative medicine.

---

## [Decision Letter]

[Editors' note: this paper was reviewed by Review Commons.]

---

## [Author Response]

Summary reviews (experiments):#1: “The overall assessment of the manuscript is consistent between reviewers. There are suggestions that would improve the manuscript, but nothing seems to put in question the main claims of the paper.”#2: “All reviewers are in agreement that this is timely, important, and potentially highly significant contribution. All comments and criticism aiming to advise authors how to improve the revised manuscript by providing more detailed descriptions of methodologies, clarifying some uncertainties regarding data analyses and presentations, and addressing minor editorial shortcomings.”#3: “I agree with the general assessment of all reviewers”Summary reviews (significance):#1: “Interest in dissecting the putative (but still unclear) roles of HERVH/LTR7 in pluripotency has been growing. This work will become a key reference and resource for those studying these elements. It provides clarity on the subfamily structure and their evolutionary relationships, and places that within the context of transcriptional regulation during preimplantation development and ESCs.”#2: “This is potentially highly significant contribution with the impact extending beyond the several areas of fundamental biological and biomedical science into the multiple areas of translational biomedicine. With the adequate context-specific editorial highlights and perspectives this paper would be of major impact and interest to the broad scientific readership and general public.”#3: “As the authors well discuss in the manuscript their analysis brings key information to the table, allowing researchers on the broad fields of hESCs biology and regenerative medicine to use that to refine the characterization of the pluripotent cells. I believe the manuscript will cater not only to transposon and genome evolution afficionados but will also be of interest to the fields of human pluripotency and regenerative medicine.”

It is clear from the comments above that all reviewers were strongly enthusiastic about the study. The reviewers also offered excellent feedback and constructive suggestions to improve the arguments and clarity of the manuscript as well as the robustness of some of the computational analyses and data/code availability. We have now carefully addressed all their suggestions, point-by-point, as detailed below. In most cases, their comments resulted in slight changes to the text to clarify certain points or add discussion points. Additionally, we present a few new bioinformatic analyses to address their more substantial suggestions. Substantive revisions can be summarized as follow:

– SOX2 reads added to heatmap and aggregate signal plot presented in supplemental figure 2 to ensure LTR7 subfamily enrichment was not an artifact of peak calling (in response to Reviewer 1).

– An additional aggregate signal plot and heat map was added as supplemental figure 8 to assess whether there was an obvious correlation between SOX2 motif presence/strength and SOX2 binding and transcriptional activity (in response to Reviewer 3).

– Additional enrichment analysis looking into the transcription of LTR7 subfamilies in multiple “primed” and “naïve” ESC lines was added as supplemental figure 10 (in response to Reviewers 1, 2, and 3).

1. Point-by-point description of the revisionsReviewer #1:1. I may be mistaken, but as far as I can tell all ESC data are from so-called 'primed' culture conditions. Given some of the differences in expression between ESCs and epiblast (as well as claims made in Wang et al; PMID: 25317556), it may be worth also evaluating the expression of the different subfamilies in naïve ESCs. It would also help to support the suggestion that 7u2 could be used as a naïve cell marker.

All reviewers agreed that we should expand the analysis of ESC to ‘naïve’ ESC (in addition to ‘primed’ ESCs). We concur and, as suggested by reviewer #1, we have now analyzed and compared RNA expression of all LTR7 subfamilies in multiple primed and naïve ESC lines, with a specific attention for the expression of the subfamilies containing the identified SOX2 motif, 7up and 7u2. The results of this new analysis are consistent with the pattern we discerned from our scRNA-seq analysis of human embryos: we find that LTR7b, y, and u2 transcripts are more abundant in naïve lines, while LTR7o, bc, up1, up2, d1, and u2 transcripts are more abundant in primed cell lines. This is in agreement with the current view in the field that naïve cells are closer to epiblast cells and that LTR7b, y and u2 mark specifically these cells. The results of this analysis are now reported as Supplemental Figure 10 and we have added the following text to integrate these findings:

“Indeed, while 7up shows expression preference in primed cell lines, 7u2 shows preference for naïve lines (Figure supplement 10).” – page 28-29

2. From Figure 3D, it appears that SOX2 binds around half of the expressed 7up loci. If this is a critical factor for expression, could it be that the remaining loci are moderately enriched, below the threshold for peak detection? It may be worth doing a parallel analysis based on enrichment levels. Or perhaps, as the authors suggest, SOX3 is the key protein.

Reviewers #1 and #3 recommended that the addition of a non-peak approach to assess SOX2 binding enrichment to 7up might be beneficial. We agree and we have now integrated an analysis of SOX2 ChIP-seq reads (in addition to peaks) into the heatmaps presented in supplemental figure 2. We find that ChIP-seq reads are evenly distributed across 7up1/2 loci (and 7u1 to a lesser extent), indicating that SOX2 binding is not driven by a few LTR loci but is a more general feature of 7up. Additionally, we note that read pileups do not seem biased toward more highly transcribed loci. Moreover, beds, bams, and related data files have now been provided in the github for others who want to replicate or expand the analysis for their own purposes.

3. Related to this, the suggestion that SOX3 is the key protein is based on its expression pattern. However, the authors also suggest that 7up elements are kept repressed by KZFPs/KAP1 at early preimplantation stages, which could prevent SOX2 from activating 7up at this point. Moreover, unlike what is claimed in the manuscript, Supplemental Figure 6C shows higher expression of SOX2 than SOX3. Whilst the jury is still out, together with the ChIP-seq data and known roles in pluripotency, the involvement of SOX2 remains likely.

There was some confusion as to whether we thought SOX2 or SOX3 was more likely to be necessary for 7up transcription. We intended to posit that both scenarios are equally likely and have adjusted wording in the manuscript accordingly. Related to this, we originally stated that SOX3 RNA was more abundant in ESC than SOX2 RNA. We have reworded this to reflect supplemental figure 6c where SOX3 RNA is more *specific* to ESC than SOX2 RNA, but not more abundant. Additional experiments would be needed to test whether the crucial factor is SOX2 or SOX3.

“In addition, we observed that both SOX2 and SOX3 are expressed in human ESCs but SOX3 was more specifically expressed in ESCs (Figure supplement 6A,C)” – page 20

4. Is the SOX2/3 motif also present in block 2a, given its close relationship with 2b? If so, can the authors speculate why the 2b motif would be so critical? Could it promote SOX2 homodimerization? Or perhaps acts cooperatively with other TFs (e.g., KLF4), as shown for motifs within RLTR9 elements in mice (PMID: 28348391).

Reviewer #1 requested further discussion on the occurrence and position of the SOX2/3 motif in block 2a and if we think homodimerization or cooperative TF binding is responsible for boosting transcriptional activity. This is an interesting speculation and we have added text in the discussion to mention it. Briefly, we do observe that block 2a also contains a SOX2/3 motif, so it is possible that homodimerization or cooperative binding explains the transcriptional activation of 7up in ESC. (5) Furthermore, 7u1 elements also have the SOX2/3 motif in block 2b, yet these elements are not transcribed. We think it likely that these subfamilies need additional TF binding sites for ESC transcription.

“This exact motif is also present in the untranscribed 7u1 subfamily. More distantly related subfamilies also have SOX2/3 binding sites elsewhere on their sequence (supp.file 4), indicating that the presence of a SOX2/3 site is not sufficient to confer transcriptional activity in ESC. Perhaps SOX2/3 cooperatively bind with other TF to uniquely activate 7up.” – page 24

5. Could the authors also briefly discuss why 7u1 elements are not expressed despite having the SOX2 motif (as well as binding, albeit less, in Figure 6B)? This would be an interesting consensus to add to the luciferase assay.

Please see response to point 4 above.

Reviewer #2:1. Please consider designing and executing targeted CRISPR/Cas9 genome editing and/or CRISPR-guided epigenetic silencing experiments to unequivocally demonstrate the role of the 8 nucleotide insert in the LTR7up sequence in maintenance of the pluripotency phenotype;

In agreement with reviewers 1 and 3 (see quoted text below), we feel such experiments are beyond the scope of the paper. There has been a slew of HERVH KD and KO experiments in recent years. As we noted in our introduction and discussion, none of these experiments have pinned down the exact role of HERVH in pluripotency. With this manuscript, we aimed to gain new insights and develop resources to eventually address this important question. But clearly it will take extensive experiments in the future to fully understand the role of HERVH in pluripotency. As reviewer 3 notes, we believe we were careful to not overstate the importance of the predicted Sox2/3 motif in the maintenance of pluripotency. Our results only show that these 8 basepairs are necessary for 7up transcription in iPSCs.

“Reviewer #2 suggests CRISPR experiments, but to me this feels outside of the scope of this paper. Others have explored, and continue to explore, the potential roles of LTR7/HERVH in pluripotency, but this is not the aim of this paper.” – Reviewer #1

“I agree with Reviewer #1 on the requirements of the CRISPR experiments. While it would be an extremely interesting approach, dissection of whether LTR7up expression is a marker or a driver of pluripotency can be an entire new manuscript, and the authors were careful not to extrapolate claims of causality related to its contribution to pluripotency.” – Reviewer #3

2. Please consider to expand sequence alignment and conservation analyses beyond the hg38 human genome reference database to include: (a) recently released "The complete sequence of a human genome" (https://doi.org/10.1101/2021.05.26.445798); (b) assessment of the conservation in human population (1000 genomes or other databases of individual human genomes; of particular interest would be databases that include genomes from individuals with different ethnic backgrounds) of the newly identified LTR7 family, particularly, candidate regulatory sequences that appear to emerge most recently (LTR7y; LTR7up); (c) assessment of the conservation in several releases of Chimpanzee genomes (may provide some important insights into conservation/variability within non-human primate species) and especially recent high-quality Bonobo genomes ("A high-quality bonobo genome refines the analysis of hominid evolution"; https://doi.org/10.1038/s41586-021-03519-x ); assessment of the conservation in most recent releases of Neanderthals and Denisovan genomes ("An ancestral recombination graph of human, Neanderthal, and Denisovan genomes"; http://advances.sciencemag.org/ ), in particular, assessment of LTR7 genomic coordinates within human-specific genomic regions lacking ancestral DNA (regions not shared with archaic hominins);

Reviewer #2 also suggested expanding our HERVH conservation analysis to (a) the newly released complete human reference genome (b) 1000 genome or other human population genome datasets (c) chimpanzee population genome datasets (d) a newly released high quality bonobo genome (e) the Denisovan and Neanderthal genomes. While we agree that these are interesting ideas, we do not believe that such analyses would reinforce or alter the major findings of the paper. The main reason being that it has been well established that HERVH was active prior to the divergence of African great apes (Goodchild et al., 1993; Izsvák et al., 2016; Mager and Freeman, 1995) and our study further confirms that at the level of individual subfamilies. Thus, it is strongly predictable that HERVH insertions have long been fixed in the human population and shared by archaic hominins as well as African hominids. In fact, our lab has previously carried out a search for polymorphic HERVH insertions in the 1000 Genomes and Simons Diversity Genome Project (Thomas et al. 2018) and did not identify a single polymorphic HERVH insertion in these datasets (unlike HERVK for which we and others have identified a few polymorphic insertions). While we cannot formally rule out rare instances of HERVH insertion polymorphisms among and between hominid species, these are unlikely to change the bigger picture of HERVH evolution depicted in our study. We have now altered the text to clarify these points.

“To date, no polymorphic HERVH insertions have been found in the human genome (Thomas et al., 2018).” – page 1

“Insertions shared at orthologous genomic position across a set of species are deemed to be ancestral to these species and thus can be inferred to be at least as old as the divergence time of these species and likely fixed within each species (Johnson, 2019).” – page 8

Likewise, expanding our analysis of the newly released “complete” human genome produced by the Telomere-to-Telomere consortium is unlikely to affect our conclusions because we do not expect many HERVH insertions to be lacking from the version of the assembly used for our analysis. This is because HERVH, like other gamma retroviruses, have a propensity to integrate into transcriptionally active genomic regions, not the heterochromatic and centromeric regions that were resolved in the T2T assembly (https://doi.org/10.1101/2021.07.12.451456).

3. Please provide detailed descriptions of the "LiftOver" methodology that was utilized in the study, including values of the minimum ratio of bases that must remap to ascertain the successful conversion between the assemblies; Please provide a complete report of the results (as the Supplemental Tables, please) for all LT7 sub-families; Specifically, please indicate what sequence conservation levels were chosen to define a successful conversion between different genomes; Please clarify whether only direct "LiftOver" or both direct and reciprocal "LiftOver" analyses were executed during the comparisons of sequence conservation between species; It is impossible to reproduce and validate the reported findings without the adequate description and clarification of these points;

Reviewer #2 requested a more detailed description of the liftover methodology. We have expanded the description in the Methods section. Briefly, liftover chain files map syntenic regions between genomes. With closely related species, most of these are 1:1 matches i.e. one region in human matches with only one orthologous chimpanzee region. Sometimes, there are multiple syntenic hits from one species to another (equally likely scenarios), often caused by duplications or deletions in one lineage. Therefore, we only use reciprocal liftover hits in our calculations i.e. one human genomic region that liftovers to one chimpanzee region that liftovers back to the same original human region (default settings of 0.95 overlap both ways). The difference between reciprocal and unidirectional liftover has been clarified in the text. All commands and files used are available in github. The corresponding data files are also now included as Supplementary files.

“Those that were successfully lifted over from human to non-human primates (syntenic regions) were then lifted over back to human. Only those that survived both liftovers (1:1 orthologous) were counted as present in non-human primates.” – page 34

4. For each figure in the main text and the supplement, please provide the corresponding data source tables (these should be provided as supplemental tables with adequate descriptions and references); This is a standard requirement for top-ranked journals which is essential for reproducibility of the reported findings;

Reviewer #2 correctly noted that some data source tables were missing (point 4) and some Supplementary files had inadequate descriptions (point 6). A description file has been added to accompany more aptly named Supplementary files. All files and commands used are also now available on github.

5. One approach to calculation of the estimated evolutionary emergence time of regulatory LTRs is linked to estimates of the divergence time between species based on estimated mutation rates. However, calculations of the mutation rate in humans as well as estimations of the divergence time between humans and chimpanzees have been the controversial subject (please see excellent discussion on these subjects in the Supplement of most recent ScinceAdvaces 2021) paper "An ancestral recombination graph of human, Neanderthal, and Denisovan genomes"; http://advances.sciencemag.org/. If the slower assumed mutation rate (0.5x10-9 per base per year or 1.25x10-8 per base per generation based on comparisons parents to offspring) is accounted for, the human-chimpanzee divergence time was estimated at 13 Mya rather than 6.5 Mya (K. Prüfer, C. de Filippo, S. Grote, F. Mafessoni, P. Korlević, M. Hajdinjak, B. Vernot, L. Skov, P. Hsieh, S. Peyrégne, D. Reher, C. Hopfe, S. Nagel, T. Maricic, Q. Fu, C. Theunert, R. Rogers, P. Skoglund, M. Chintalapati, M. Dannemann, B. J. Nelson, F. M. Key, P. Rudan, Ž. Kućan, I. Gušić, L. V. Golovanova, V. B. Doronichev, N. Patterson, D. Reich, E. E. Eichler, M. Slatkin, M. H. Schierup, A. M. Andrés, J. Kelso, M. Meyer, S. Pääbo, A high-coverage Neandertal genome from Vindija Cave in Croatia. Science 358, 655-658 2017).This approach was successfully implemented in most recent ScienceAdvances (2021) paper "An ancestral recombination graph of human, Neanderthal, and Denisovan genomes"; http://advances.sciencemag.org/. It would be of interest to determine how the proposed evolutionary timeline of the emergence of distinct LTR7 regulatory sequences would be affected when the approach based on slower assumed mutation rate is implemented.

Reviewer #2 also expressed interest in how our aging analysis would change with different divergence time calculations. We appreciate that divergence times between these lineages are estimates that are often contested and revised. However, using different divergence times would only affect the dating of subfamily expansion and would not alter the sequence of amplification we infer from our analysis because our dating is based on comparative genomics and an analysis of shared insertions (liftover) rather than a molecular clock.

6. Please provide titles and short descriptions for individual Supplementary files with references to figures, tables, or other relevant contents of the main text; This is the standard requirement for peer-reviewed publication which is essential for reproducibility of the reported findings

Please see above response to point 4.

Reviewer #3:1. On the mapping of ChIP-seq reads (and other NGS datasets used), what were the parameters used for dealing with multi-mappers? Some parameters can cause an artificial "inflation" of the inferred signal over abundant repeats. E.g. allowing mismatches + report all would erroneously subscribe an active status to several LTR copies when perhaps in reality all signal is coming from one or few individual copies. This can of course affect the conclusions: if many phylogenetically close LTRs have "active" signals this likely points towards a sequence dependent effect (e.g. a reader TF as in this manuscript), however if most "active" signal derives from a few copies this can point towards a position/context effect (e.g. proximity to other regulatory regions). I like the approach the authors took of extrapolating measured signal of flanking sequences that partially overlap with the LTR annotation, but how does it compare with unique mapping with no mismatches?

Reviewer #3 requested a more detailed description of the ChIP-seq mapping methodology, especially regarding multi-mappers. This is an especially important point with relatively young repetitive elements. Our use of a phred score of 33 as a mappability cutoff is already quite stringent. However, we repeated our analysis of SOX2 binding LTR7 subfamilies (Figure 6B) using only uniquely mappable reads (see Author response image 1). While there is less overall coverage, no subfamilies are uniquely affected and the overall picture remains the same. We have added text to the methods to reflect these new data. For our ChIP-seq enrichment analyses, we chose to keep our initial cutoff of Phred >33 as the gapped alignments may disturb the Poisson distribution of piled reads.

**Author response image 1. sa2fig1:** 

“This analysis was repeated with exclusively uniquely mapping reads which did not affect the outcome of analysis (Data not shown).” – page 38

2. Maybe a metaplot + hierarchical heatmap for the stacked members of each of the 8 LTR subfamilies showing selected regulatory signals depicted on Figure 3? I like figure 3A, but its binary nature doesn't allow us to get a sense of the quantitative distribution of these signals over the many members of each subfamily. The other extreme are the heatmaps on Figure 3 B-C-D which are quantitative but suppress the information about the variability. Perhaps the chosen formats relate to the limitations of the NGS mapping (see above).

Please see response to reviewer #1 comment #2 above.

3. Relates to the correlation between SOX2/3 binding sites and transcription in ES. If using quantitative approaches to the regulatory data as discussed above, can the authors correlate strength of SOX2/3 motifs with activity? Would this expose another layer of variability within active subfamilies?

Reviewer #3 suggested performing an additional analysis to correlate the presence of the SOX2/3 binding site in block 2b and ESC expression. To probe this relationship, we took all 7up 5’ and solo LTRs from the human genome and sorted them based on their HOMER SOX3 pvalue hit for this motif. We then layered on SOX2 ESC ChIP-seq onto these coordinates. We found that there is no correlation between SOX2 binding and the strength of the SOX3 binding motif prediction. The data have been added to the supplement (Supplemental Figure 8) and the text has been modified to include the results.

“Additionally, the strength of the motif match does not correlate with SOX2 binding or transcription within 7up (Figure supplement 8). Perhaps SOX2/3 cooperatively bind with other TF to uniquely activate 7up.” – page 24

Minor revisions:Figure 1A and 1B legends were swapped.

This has been corrected.

Several typos have been corrected*.*

Figure 2A was referred to as 3A several times in the text. Figure 4C was also referred to as 4D.

This has been remedied.

Several phrasing discrepancies were fixed.

The text and Supplementary files were unclear on if they were from previously annotated or remasked data sources. This resulted in seeming discrepancies in insertion counts between the text and supplement.

The text, Supplementary files, and readmes have been changed to mark more strongly designate their source. In the example listed by reviewer #2. The supp table lists 113 full and 524 solo LTR7b (637 total – same as text) counts from “4.0.5 repeat Library 20140131”. The labeling of this section has been improved.